# Sensitivity of the tropical stratospheric ozone response to the solar rotational cycle in observations and chemistry-climate model simulations

Rémi Thiéblemont[1], Marion Marchand[1], Slimane Bekki[1], Sébastien Bossay[1], Franck Lefèvre[1], Mustapha Meftah[1], Alain Hauchecorne[1]

[1]Laboratoire, Milieux, Observations Spatiales, Institut Pierre Simon Laplace, CNRS, Paris, France.

*Correspondence to*: Rémi Thiéblemont (remi.thieblemont@latmos.ipsl.fr)

**Abstract.** The tropical stratospheric ozone response to solar UV variations associated with the rotational cycle (~27 days) is analysed using MLS satellite observations and numerical simulations from the LMDz-Reprobus chemistry-climate model. The model is used in two configurations, as a chemistry-transport model (CTM) where dynamics are nudged toward ERA-Interim reanalysis and as a chemistry-climate model (free-running) (CCM). An ensemble of five 17year simulations (1991-2007) is performed with the CCM. All simulations are forced by reconstructed time-varying solar spectral irradiance from the Naval Research Laboratory Solar Spectral Irradiance model. We first examine the ozone response to the solar rotational cycle during two 3year periods which correspond to the declining phases of solar cycle 22 (10/1991-09/1994) and solar cycle 23 (09/2004-08/2007) when the satellite ozone observations of the two Microwave Limb Sounders (UARS MLS and Aura MLS) are available. In the observations, during the first period, ozone and UV flux are found to be correlated between about 10 and 1 hPa with a maximum of 0.29 at ~5 hPa; the ozone sensitivity (% change in ozone for 1% change in UV) peaks at ~0.4. Correlation during the second period is weaker and has a peak ozone sensitivity of only 0.2, possibly due to the fact that the solar forcing is weaker during that period. The CTM simulation reproduces most of these observed features, including the differences between the two periods. The CCM ensemble mean results comparatively show much smaller differences between the two periods, suggesting that the amplitude of the rotational ozone signal estimated from MLS observations or the CTM simulation is strongly influenced by other (non-solar) sources of variability, notably dynamics. The analysis of the ensemble of CCM simulations shows that the estimation of the ensemble mean ozone sensitivity does not vary significantly either with the amplitude of the solar rotational fluctuations, or with the size of the time window used for the ozone sensitivity retrieval. In contrast, the uncertainty of the ozone sensitivity estimate significantly increases during periods of decreasing amplitude of solar rotational fluctuations (also coinciding with minimum phases of the solar cycle), and for decreasing size of the time window analysis. We found that a minimum of 3year and 10year time window is needed for the 1σ uncertainty to drop below 50% and 20%, respectively. These uncertainty sources may explain some of the discrepancies found in previous estimates of the ozone response to the solar rotational cycle.

# 1 Introduction

The thermal structure and the composition of the middle atmosphere are sensitive to fluctuations in the incoming solar radiation, which in turn can affect the Earth's surface climate variability (Gray et al., 2010). These solar variations are dominated by the 11year solar magnetic activity cycle and the solar rotational cycle, also called 27day solar cycle. Changes in total solar irradiance (TSI) over an 11year solar cycle are typically lower than 0.1%, that correspond to 1 W m$^{-2}$ change for a reference value of $1360.8 \pm 0.5$ W m$^{-2}$ (Kopp and Lean, 2011). Such small variations in the total energy input are not expected to have a significant impact on climate, compared for instance to the variations of anthropogenic origin, and thus air-sea coupling mechanisms have been proposed that act to amplify the small solar initial perturbations (e.g. Meehl et al., 2008). Another possible amplification mechanism, also known as "top-down" (Kodera and Kuroda, 2002), operates through changes in the spectral solar irradiance (SSI) - in particular in the ultraviolet (UV) range - that directly modulate the stratospheric temperatures and ozone concentrations. These perturbations induce dynamical changes in the stratosphere, which may in turn affect the tropospheric circulation through stratosphere-troposphere couplings (e.g. Gerber et al., 2012). A thorough understanding and accurate quantification of the UV variability effect on the middle stratosphere ozone are thus necessary.

Solar irradiance fluctuations strongly depend on the wavelength range and their relative amplitudes tend to increase sharply with decreasing wavelengths (Lean, 2000). In the UV range, the variability over the course of the 11year solar cycle is of about 8% at 200 nm. Several observational and modeling studies have examined the impact of 11year UV variability on stratospheric ozone and temperature (e.g. Hood, 2004; Soukharev and Hood, 2006; Randel and Wu, 2007; Austin et al., 2008; Remsberg et al., *2008*; Gray et al., 2009; Remsberg, 2014; Dhomse et al., 2016). These studies found a change associated with 11year solar cycle in the range of 2 to 5% in ozone mixing ratio, which maximizes near 40 km. Maycock et al. (2016) recently compared the ozone 11year solar cycle signal of several different satellite records and found substantial differences. One inherent issue of the observational investigation of the 11year cycle ozone response is the fact that only three complete periods of the 11year solar cycle have been covered by satellite observations so far. Furthermore, the life span of a single satellite instrument is generally shorter than (comparable to in some cases, e.g. TIMED SABER, ENVISAT MIPAS, ENVISAT GOMOS, Aura MLS) one solar cycle and instrumental biases between different ozone profile data sets complicate statistical analysis of decadal variations (Fioletov, 2009; Dhomse et al., 2016). In this regard, a suitable alternative for understanding better the direct effect is to examine the ozone response on Sun's rotational timescale (i.e. about 27 days). Although the irradiance fluctuations during the rotational cycle are on average smaller than during the 11year solar cycle, there are many more rotational cycles than 11year cycles, improving considerably the statistics.

A number of observational studies has been carried out to determine the effects of the solar rotational cycle on stratospheric ozone, generally at low-latitudes (i.e. tropical region) based on the analysis of satellite observations (e.g. Hood, 1986; Eckman, 1986b; Keating et al., 1985; 1987; Hood et al., 1991; Fleming et al., 1995; Hood and Zhou, 1998; 1999; Fioletov, 2009; Dikty et al., 2010). These studies have shown that the sensitivity of tropical ozone to the solar rotational cycle maximizes at about 40 km (or ~3 hPa) and varies from 0.2 to 0.6% for a 1% change in solar UV radiation index, typically taken as the irradiance

at the 205 nm wavelength. It was further shown that the phase lag of the tropical stratospheric ozone response varies with the altitude. The phase lag vertical profile between the ozone response and the solar forcing was found to be negligible at about 40 km and gradually increasing/decreasing, below/above that altitude. The phase lag was estimated to be approximately 4 days at 30 km and -2 days at 50 km (e.g. Hood, 1999; and references therein).

Simulations with numerical models of various complexities have been performed to understand the influence of the rotational cycle on ozone variability. One-dimensional photochemical-radiative model experiments (e.g. Hood, 1986; Eckman, 1986a; Brasseur et al., 1987) allowed identifying the importance of temperature/ozone couplings and reproducing the gross features of the observed ozone response. In particular, they found that the negative phase lag between the solar forcing and the ozone response in the upper stratosphere originated from the strong influence of the temperature feedback on ozone response through

the temperature dependent chemical reactions (Brasseur et al., 1987). They however noticed that including the solar induced temperature changes alone was not sufficient to adequately reproduce the observed magnitude and phase lag of the ozone response and suggested that atmospheric dynamical variability – which is not simulated in 1-D models - may also have a sizeable influence (Hood, 1986; Brasseur et al., 1987). The latter issue has later been addressed with two-dimensional models which revealed better agreement with observations (Brasseur, 1993; Fleming et al., 1995; Chen et al., 1997). Fleming et al.

(1995) further stressed the increasing importance with height of the solar-modulated HOx chemistry on the ozone response above 45 km. In the upper stratosphere and mesosphere, enhancement of HOx through photolysis of water vapour in Lyman-alpha line associated with an increasing solar irradiance contribute to destroy ozone. Above ~65 km and at zero-lag, the latter mechanism dominates over ozone production (i.e. by photolysis of oxygen) leading to a negative ozone-solar irradiance correlation. In the upper stratosphere and lower mesosphere (below 65 km), although ozone production dominates, increasing

HOx at zero-lag contributes to the negative lag of the ozone response (Rozanov et al., 2006).

Using a large ensemble (nine 1year long runs) of chemistry-climate model (CCM) simulations, Rozanov et al. (2006) found that the ensemble mean ozone sensitivity to the solar rotational irradiance changes was in very good agreement with observational data. They however pointed out – despite an identical solar forcing for each experiment - a large scatter in maximum ozone sensitivities that could vary by a factor of almost 10 between the two most distant ensemble members. A

large variability in ozone sensitivity was similarly found in an ensemble of three transient CCM simulations (1960-2005) (Austin et al., 2007). Bossay et al. (2015) analysed satellite observations of two periods of 3 years during the declining phases of cycles 22 and 23 (i.e. 1991-1994 and 2004-2007) and found that the solar rotational signal in stratospheric ozone time series strongly varies from one year to another. These results suggest that the background dynamical state and variability of the atmosphere contribute to masking the solar rotational signal in ozone (Gruzdev et al., 2009).

In addition to the dynamics, the intensity of the solar forcing naturally modulates the solar rotational signal in ozone. When the solar rotational fluctuations are well marked with large amplitudes, notably around the maxima of 11year cycles (e.g. Rottman et al., 2004), ozone response and correlation are expected to be the largest. This has been supported by observational (e.g. Hood, 1986; Zhou et al., 2000; Fioletov, 2009; Ditky et al., 2010) as well as modeling (Kubin et al., 2011) studies which demonstrated a better identification of the ozone signal associated with enhanced rotational forcing fluctuations. This

relationship has however been challenged by contradictory results. Hood and Zhou (1998) analysed UARS MLS ozone data for the 1991-1994 period and found a correlation two times stronger during the last half of the period, i.e. when the rotational forcing fluctuations are reduced. They suggested that it might have been the result of an artefact of either instrumental or geometric (local time coverage) origin that may have affected the earliest part of the UARS MLS ozone record more than the later part. In their recent observational study which compared the declining phases of cycle 22 and cycle 23, Bossay et al. (2015) further showed that even though the amplitude of solar rotational fluctuations of the 205 nm flux was by far the largest during the first year of both periods, the correlation with tropical ozone was found to be maximum the subsequent years.

The ozone sensitivity response to the solar rotational forcing has also been suggested to vary with the intensity of the forcing. We recall that the "sensitivity" is a quantity expressed as % changes in ozone (or any other variable of interest) per % change of the forcing (here specifically solar). Hence, the sensitivity is normalized by the amplitude of the forcing and may not be expected to change strongly with the amplitude of the forcing, or at least not as much as the absolute amplitude of the ozone response which directly depends on the amplitude of the forcing. Gruzdev et al. (2009) used an idealized solar rotational forcing in their model (prescribed as a sinusoidal 27day oscillation) and found a significant reduction of the ozone sensitivity when applying an enhanced solar forcing amplitude (3 times the standard amplitude). Reciprocally, in the CCM experiments of Kubin et al. (2011), the ozone sensitivity seemed to be enhanced during periods of weak 27 day cycles. Finally, the observational study of Bossay et al. (2015) also hints at an opposite relationship between the solar rotational irradiance fluctuations and the ozone sensitivity. Given the strong influence of the dynamical background state on the variability of estimated ozone sensitivity and the rather shortness of the considered time windows of analysis, they recognized that it was not possible to conclude to a systematic effect. All these results thus highlight the uncertainty regarding the influence of the forcing intensity on ozone sensitivity and on the length of the time window required for an accurate and robust estimation of the ozone rotational signal.

In the present study, we examine the sensitivity of the tropical stratospheric ozone response to the rotational cycle by comparing satellite observations and chemistry climate model experiments to understand better the origin of the discrepancies - and sometime contradictory results - in the estimation of the ozone response to the solar rotational cycle found in previous studies. As a first step, we follow up on the case study of Bossay et al. [2015] and make use of observations and modelling results comparison to provide a detailed picture of the ozone response to the solar rotational cycle during the declining phases of cycle 22 and cycle 23. We particularly aim to better understand the strong differences in the ozone response to solar rotational cycle found between the two periods. Two configurations of the LMDz-Reprobus chemistry climate model simulations are used, with specified dynamics (i.e. Chemistry Transport Model, or CTM) and in its free running mode (CCM). In the CTM configuration, temperature and wind fields calculated by the model are relaxed towards meteorological analysis; the dynamics is expected to be rather close to the reality, allowing direct comparisons with satellite observations for evaluating model chemical processes and its relevance to our study. In the CCM configuration, an ensemble of simulation is performed. Comparing the CCM ensemble results to CTM and observations during the declining phases of cycle 22 and cycle 23 allows to understand better the effect of internal dynamical variability on the ozone response. As a second step, we take advantage of

the ensemble of CCM simulations and its large statistics to (i) assess the influence of the solar cycle phase on the ozone sensitivity to the rotational cycle and (ii) quantify the time window required for a robust estimation of the ozone sensitivity. Observational datasets, and model configurations and simulations are described in section 2. Section 3 presents comparisons between satellite observations and model (CTM and CCM) simulations of the ozone response to the solar rotational cycle.

Section 4 focuses on CCM results to examine the influence of (i) the solar activity fluctuations and (ii) the length of the time window in the estimation of the ozone sensitivity to the solar rotational cycle. The main findings are summarized in section 5. Note that for the sake of simplicity, the first period (10/1991-09/1994) during cycle 22 will be referred thereafter as 1991-94 period and the second period (09/2004-08/2007) during cycle 23 will be referred as 2004-07 period.

## 2 Data and model description

### 2.1 The 205 nm solar flux (or F205)

The solar proxy used in regressions analyses is the UV solar irradiance at 205 nm. This wavelength is chosen because it is important for the ozone chemical budget throughout the stratosphere. The 205 nm wavelength is included in the Herzberg continuum region (200-242 nm) that is positioned between two strong absorption bands: the Schumann–Runge band of molecular oxygen and the Hartley band of ozone (Brasseur and Solomon, 2005). In the Herzberg continuum, atmospheric

absorption is relatively low and hence solar UV radiation penetrates deeply in the atmosphere, down to the lower stratosphere, where it photolysis molecular oxygen ($O_2$) to produce $O_3$. The 205 nm flux, called thereafter F205, has been commonly used in previous studies because it is a very good proxy for characterizing solar variability in the UV domain.

In our study, we use the solar spectral irradiance provided by the Naval Research Laboratory Solar Spectral Irradiance (NRLSSI) model version 1 (Lean, 2000; Wang et al., 2005). NRLSSI is an empirical model which aims to reconstruct long-

term SSI over the wavelength domain 120-100,000 nm. It uses historical estimates of faculae brightening and sunspot darkening to extend in time wavelength-dependent parameterizations of SSI derived from satellite measurements and model. Shortwards of 400 nm, the SSI is derived from UARS/SOLSTICE observations (Rottman et al., 2001) through a multiple regression analysis with respect to a SOLSTICE reference spectrum. The regression analysis includes a facular brightening and a sunspot darkening time-dependent term. Above 400 nm the SSI is reconstructed by adding the irradiance changes caused

by the presence and the characteristics of faculae and sunspots (see Lean (2000) for details) to a quiet Sun intensity spectrum, i.e., defined by the absence of faculae and sunspots. The intensity spectrum of the quiet Sun is a composite compiled from space-based observations made by UARS/SOLSTICE (120-401 nm) and SOLSPEC/ATLAS-1 (401-874 nm) (Thuillier et al., 1998), and a theoretical spectrum at longer wavelengths (Kurucz, 1991).

### 2.2 Microwave Limb Sounder ozone satellite observations

We use the stratospheric ozone measurements from the two Microwave Limb Sounder (MLS) instruments on-board UARS (cycle 22) and Aura (cycle 23).

UARS MLS was launched on 12 September 1991, into a 57° inclination and a 585 km altitude orbit and was operational until 1994. Waters (1989; 1993) describe in detail the microwave limb-sounding technique. We used the version 5 UARS MLS dataset described Livesey et al., (2003). The ozone retrieval is based on 205 GHz radiances, provided onto 13 pressure levels in the range 100-1 hPa (100, 68.1, 46.4, 31.6, 21.5, 14.7, 10, 6.8, 4.6, 3.2, 2.2, 1.5 and 1 hPa) and has an average vertical resolution of 4 km in the stratosphere. The typical $1\sigma$ precision for ozone mixing ratio measurements is ~0.3 ppmv between 68 and 1 hPa. As shown in Hood and Zhou (1998), an artificial 36day periodicity, caused by the UARS yaw manoeuvre cycle (Froidevaux et al., 1994), is seen in zonally averaged UARS MLS data at all latitudes and increasing with height. To remove this artefact, Hood and Zhou (1998) suggested restricting zonal averaging ozone profiles to daytime measurement near a single local time. They however recognized that the ratio of daytime measurements per day would be too low (around 30%), resulting in very large sampling errors and time gaps in the zonal averages. Furthermore, ozone diurnal cycle becomes important in the upper stratosphere so that the results may be affected by the imbalance in daytime and night-time measurements used to construct daily time series. This issue will be discussed in section 3.2.

Aura MLS was launched on 15 July 2004 into a sun-synchronous near-polar orbit around 705 km. Detailed information on the Aura MLS instrument is given in Waters et al. (2006). In brief, Aura MLS observes a large suite of atmospheric parameters by measuring millimeter and submillimeter-wavelength thermal emission from Earth's limb with seven radiometers covering five broad spectral regions (118, 190, 240, 640 GHz and 2.5 THz). The "standard product" of ozone is retrieved from radiance measurement near the 240 GHz. Here, we used version 4.2 of the Aura MLS ozone product (Livesey et al., 2017). The Aura MLS fields of view point forward in the direction of orbital motion and vertically scan the limb in the orbit plane, resulting in a data coverage from 82°N to 82°S latitude on every orbit. Aura MLS provides continuous daily sampling of both polar regions without temporal gaps from yaw maneuvers that occurred with UARS MLS. The Aura MLS limb scans are synchronized to the Aura orbit, with 240 scans per orbit at essentially fixed latitudes. This results in about 3500 scans per day, with an along-track separation between adjacent retrieved profiles of 1.5° great circle angle. Ozone profiles are provided onto 25 pressure levels in the range 100-1 hPa (100, 82.5, 68.1, 56.2, 46.4, 38.3, 31.6, 26.1, 21.5, 17.8, 14.7, 12.1, 10, 8.2, 6.8, 5.6, 4.6, 3.8, 3.2, 2.6, 2.1, 1.8, 1.5, 1.2 and 1 hPa) with an average vertical resolution of 3 km in the stratosphere. The $1\sigma$ precision for ozone mixing ratio measurements is about 0.1 to 0.3 from 46 hPa to 0.5 hPa.

For our study, daily stratospheric ozone profiles averaged over the tropical band [20°S,20°N] are used. Among the 1095 days of each period, 121 and 38 days of ozone data are missing for the period 1991-94 and 2004-07, respectively. For each height level of the vertical profile, the outliers of the corresponding ozone time series are removed by excluding data which take absolute values beyond 2 standard deviations of the deaseasonnalized time series. After removing outlier values, 85% and 93% of the 1095day ozone time series of the periods 1991-94 and 2004-07, respectively, are kept for the analysis.

## 2.3 The LMDz-Reprobus model

The LMDz-Reprobus model is a Chemistry-Climate Model resulting from the coupling between the extended version of the General Circulation Model LMDZ5 (Sadourny and Laval, 1984; Le Treut et al., 1994; 1998; Lott et al., 2005; Hourdin et al.,

2006; 2013) and the chemistry module of the Reprobus stratospheric chemistry-transport model (Lefèvre et al., 1994; Lefèvre et al., 1998). LMDZ was developed at the Laboratoire de Météorologie Dynamique (LMD). The dynamical part of the code is based on a finite-difference formulation of the primitive equations of meteorology (Sadourny and Laval, 1984). The model uses a classical hybrid $\sigma$-P coordinate in the vertical, has 39 vertical levels and a lid-height at ~70 km. The model vertical resolution slowly decreases with height. In the middle and upper stratosphere (30-50 km or ~10-1 hPa) - focus of our study – the model vertical resolution reaches 3 km which is similar to the vertical resolution of UARS MLS and Aura MLS measurements in this altitude range. The model is integrated with a horizontal resolution of 3.75° in longitude and 1.9° in latitude. The equations are discretized on a staggered and stretched latitude-longitude Arakawa-C grid.

The Reprobus chemistry model (Jourdain et al., 2008; Marchand et al., 2012) calculates the chemical evolution of 55 atmospheric species and includes a comprehensive description of the stratospheric chemistry ($O_x$, $NO_x$, $HO_x$, $ClO_x$, $BrO_x$ and $CHO_x$). It uses 160 gas-phase reactions and 6 heterogeneous reactions on sulfuric acid aerosols and PSCs. Absorption cross-sections and kinetics data are based on the 2011 Jet Propulsion Laboratory (JPL) evaluation (Sander et al., 2011). In the troposphere, where the chemistry is not explicitly treated, the model is relaxed towards a monthly varying climatology (annual cycle) of $O_3$, CO and $NO_x$ computed by the TOMCAT chemical-transport model (Law et al., 1998; Savage et al., 2004).

The solar component of the radiative scheme of LMDZ5 is based on an improved version of the two bands scheme developed by Fouquart and Bonnel (1980) and the thermal infrared part of the radiative code is taken from Morcrette et al. (1986). While this scheme is crude, note that the thermal component of the solar forcing (e.g. changes in net heating from solar changes only, keeping chemical composition unchanged) does not exhibit a dependency on wavelength as strong as photolysis component of the solar forcing. Nonetheless, the use of a simple two bands radiation code tends to underestimate the temperature response when compare to other radiations models with the same solar irradiance fluctuations (CCMVal, 2010; Forster et al., 2011). The radiative scheme takes into account the radiative active species $H_2O$, $CO_2$, $O_3$, $O_2$, $N_2O$, $CH_4$, CFC-11 and CFC-12.

The photolysis rates used in Reprobus are pre-calculated off-line with the Tropospheric and Ultraviolet Visible (TUV) model (Madronich and Flocke, 1999; Sukhodolov et al., 2016) and then tabulated in a look-up table for 101 altitudes, 7 total ozone columns and 27 solar zenith angles. TUV calculates in spherical geometry the actinic flux, scattering and absorption through the atmosphere by the multi-stream discrete ordinate method of (Stamnes et al., 1988). The spectral domain extends from 116 to 850 nm. Calculations of photolysis rate are performed on a 1 nm wavelength grid, except in the regions relevant for solar cycles (rotational and 11-year solar cycles). In these spectral regions, the resolution is largely increased to accurately describe the spectral features in the solar flux or in the absorption cross-sections: the wavelength resolution increases up to 0.01 nm in the Schumann-Runge bands of $O_2$. At this resolution, the absorption by $O_2$ can be considered to be treated line-by-line. Moreover, the temperature dependent polynomial coefficient determined by Minschwaner et al. (1992) is used. The temperature dependence of absorption cross-sections is calculated off-line in TUV using the US standard atmosphere. The albedo considered for the computation of photolysis rates is set to a globally average value of 0.3 with solar zenith angle varying from 0 to 95°. For each sunlit grid point, the actual photolysis rates used by LMDz-Reprobus are then interpolated in the table according to those parameters (solar zenith angle, ozone column, altitude). The solar rotational cycle forcing is taken

into account by using daily photolysis rates calculated by TUV in the photochemistry module of LMDz-Reprobus. A separate photolysis look-up table is calculated every day using the daily NRLSSI as solar input. Note however that the direct effect on heating rates generated by UV variations associated with the 27day rotational cycle is neglected: i.e. daily changes in the spectral irradiance are not considered in the CCM radiative scheme. As a consequence, part of the thermal and dynamical responses to the 27day rotational cycle and hence their effect on ozone (through transport and temperature dependent chemical reactions, as described above) are missing. The impact of this approximation on our results will be discussed thereafter (sections 3 and 5).

LMDz-Reprobus is used in two configurations. The first one is the free-running model configuration (i.e. CCM) that accounts for all the interactions between chemistry, dynamics and radiation. LMDz-Reprobus is additionally used in its nudged version (i.e. CTM) where transport and dynamics are nudged towards temperatures and winds from the 6 hourly ECMWF model outputs (ERA-interim (Dee et al., 2011)). As the dynamics is specified and is close to observations, the CTM configuration allows a fair comparison with MLS observations. The CTM configuration is used over the two 3year periods of MLS ozone measurements, as analysed in Bossay et al. (2015). In the CCM configuration, we perform an ensemble of five simulations of 17 years each (from 1991 to 2007). As for the observations, we use the daily stratospheric ozone profiles averaged over the tropical band [20°S,20°N].

## 3 Ozone response to the solar rotational cycle during the declining phase of solar cycles 22 and 23

In this section, we analyse the ozone response to the solar rotational cycle over the declining phase of solar cycles 22 and 23 in the observations and in the CTM and CCM model simulations. The analysis presented here follows up on Bossay et al. (2015) observational study. In particular, we aim to assess the model performances, understand better the differences in the results between the two solar declining phase periods and highlight the importance of internal dynamical variability.

### 3.1 The rotational cycle in UV irradiance

Figure 1 shows the solar UV variability represented by F205 from 1985 to 2008 with the two periods of interest highlighted in red which correspond to the declining phase of solar cycles 22 and 23. F205 is a good indicator of the NRLSSI solar forcing prescribed in CTM and CCM simulations. Thereafter, F205 is used as the UV index in the regression analysis of the solar signal in stratospheric ozone from MLS observations and model simulations.

The Fast Fourier Transform (FFT) power spectra of the two F205 declining periods time series are shown on Fig. 2 (top panel). For both periods, the high frequency spectrum is dominated by a strong peak centred around 27 days corresponding to the main solar rotational periodicity. The broadness of the peaks indicates that the solar rotational cycle is not regular and covers a rather wide frequency domain. A small secondary peak is also found at ~13.5 days which corresponds to the first harmonic of the rotational cycle and to the presence on the Sun surface of two sunspots which rotate with the same period but are separated by about 180° in longitude (e.g. Bai, 2003; Zhang et al., 2007). The time-resolved power spectral density derived

from the continuous wavelet transforms (CWT, (Torrence and Compo, 1998)) of the two F205 time series are shown on Fig. 2 (bottom panels). CWT spectral analysis reveals that the solar rotational component strongly varies in time for both declining periods. Overall, the rotational component decreases over the declining solar activity periods and even can sporadically disappear for several months (e.g. late boreal summer 1993, spring 2006 and winter/spring 2006/2007). In addition, the solar rotational fluctuations are stronger during the first period than the second period (see Fig. 1 and 2). As the solar rotational forcing is stronger during the first period, one might expect the solar signal in ozone to be clearer.

**3.2 Observed and modelled ozone response to the rotational cycle**

We first examine potential rotational periodicities in upper stratospheric tropical ozone by carrying out a spectral analysis of daily stratospheric ozone time series averaged over the tropical band [20°S-20°N]. Figure 3 shows the normalized Lomb-Scargle periodograms (well adapted for non-continuous series, Lomb (1976); Scargle (1982)) of tropical stratospheric ozone from observations (Figs. 3a,d), CTM (Figs. 3b,e) and CCM results (Figs. 3c,f), calculated for the declining period of cycle 22 (Figs. 3a,b,c) and cycle 23 (Figs. 3d,e,f). Periodograms are shown for the 3.2 hPa (~40 km) pressure level, close to the altitude where the ozone solar signal maximizes (Hood, 1986).

The two periodograms of MLS ozone measurements (Fig. 3a and Fig. 3d) reveal no prominent peak in the range of the 20-30 days period, suggesting an absence of a solar rotational signal in ozone. More prominent peaks are found at longer periods although they are not consistent between the two periods. The large peak found at the 35day period for 1991-94 corresponds to the yaw-maneuver period of the MLS instrument as described previously (Froidevaux et al., 1994; Hood and Zhou, 1998). Similarly to observations, the periodograms of CTM results (Fig. 3b and Fig. 3e) does also not exhibit a distinctive solar rotational peak; there are some minor peaks between 20 and 30 days and their amplitudes are smaller in 2004-07 than in 1991-94. The analysis has been repeated at lower pressure-height levels (e.g. 10 hPa, not shown) and led to the same conclusions. Overall, the raw power spectrum analysis of observations and CTM results in the middle and upper tropical stratosphere does not allow identifying an ozone signal associated with the solar forcing fluctuations at rotational timescales for the two periods considered here.

In contrast, the periodogram averaged over the five CCM simulations exhibits a distinctive peak centred at 27 days for 1991-94 (Fig. 3c). For 2004-07, the peak is centred at 25 days (Fig. 3f). The peak is also less pronounced than in 1991-94, presumably because of the smaller amplitude of solar rotational fluctuations and hence model forcing in 2004-07 (see Fig. 2). However, the $2\sigma$ standard deviation (i.e. spread of the ensemble simulations) associated with these peaks is very large, indicating the presence of a strong high frequency (periods < 50 days) natural variability in ozone in this region. This illustrates the difficulty in detecting solar rotational signals in the observations, as well as in a single ensemble member over these 3year periods. Note that we additionally computed periodograms in observations during solar maximum phases (i.e. 2012-2015) where 27day fluctuations in the solar forcing are stronger than during the declining phase (not shown). The results were however similar and no clear peak at 27 days could be identified. Hence, the absence of a distinctive rotational signal suggests the presence of

strong and rather random ozone variability of non-solar origin which makes the ozone rotational signal very difficult to detect and estimate.

We further examine the relationship between stratospheric ozone and solar rotational cycle by performing cross-spectrum analysis between stratospheric ozone and F205. Despite the absence of a solar rotational peak in the ozone power spectrum derived from observations and CTM results, cross-spectrum analysis should help identifying coherent variability modes between the solar forcing and tropical ozone. Figure 4 presents the vertical profile of the magnitude-squared coherence (hereinafter referred as coherence) between F205 and tropical stratospheric ozone from MLS observations (a and d), CTM model results (b and e) and CCM model results (c and f).

A strong and statistically significant coherence is found for UARS MLS (1991-94) between 20 and 28 days and between about 10 and 1 hPa with a maximum of about 0.7 at the 22day period around 6 hPa. In contrast, the coherence for Aura MLS (2004-07) is generally weaker with only a small patch of significant coherence at the 90% confidence level. The coherence fields from the CTM results resemble those of the observations and reproduce the main features during the two periods. The main difference between observed and CTM signals is that the coherence patch extends farther to lower levels in the CTM (down to 15 hPa) and covers longer periods (20 to 33 days at ~10 hPa). For the 1991-94 period, the CTM results also overestimate the coherence around 13.5 days compared to observations.

The general features in the coherence fields from CCM results are also consistent with those of the observations. However, the area of statistical significant coherence around the 27day period is wider in the CCM results. In addition, the coherence patch does not extend as low as the CTM results. The differences observed between the MLS coherence fields of the two periods are also reasonably well reproduced in the CCM coherence results. As for the CTM fields in 1991-94, CCM results reveal a secondary area of significant signal centred at about 13.5day period and extends almost throughout the stratosphere. For 2004-07, there is no significant signal around 13-14 days in all the coherence fields. This is consistent with the UV forcing (Fig. 2) exhibiting a stronger 13.5day period component in 1991-94.

To further test the robustness of the coherence signal, we perform an additional CCM simulation for the period 1991-1997 where the solar forcing is kept constant by using fixed (i.e. climatological) photolysis rates during the model simulation. Results are shown on Fig. 5. Below 15 hPa, the different experiments show no significant coherence between ozone and solar flux. Between 15 and 1 hPa, all forced experiments (black lines) reveal a similar and significant coherence signal while for the constant solar forcing experiment (red line), the coherence is weak and within the range of randomness. The absence of significant coherence found in the constant solar experiment confirms that the coherence found between F205 and stratospheric ozone is not fortuitous and primarily originates from photolysis processes. We can also note that the reduced coherence for 2004-07 may be expected because the solar rotational fluctuations are smaller during that period compared to 1991-94 (Fig. 2). To summarize these first steps in our analysis, we find that, despite the weak magnitude of the signal, the upper stratosphere tropical ozone concentration fluctuates coherently with UV variability at solar rotational timescales.

To focus on periodicities relevant to the solar rotational cycle (13.5 and 27 days), all the time series are now filtered using the digital filter that has been commonly used in previous solar rotational studies (e.g; Hood, 1986; Chandra, 1986; Keating et al.,

1987; Hood and Zhou, 1998 and Zhou et al., 2000). The filtering procedure consists of smoothing data with a 7day running mean which removes short-term fluctuations. Linear trend and mean value are also removed from these smoothed time series. Finally, a 35day running mean is subtracted from the data, removing long-term fluctuations (e.g. seasonal, semi-annual, annual and QBO variations). The overall procedure is more or less equivalent to a 7-35 days band-pass filter in the frequency domain.

The vertical extent and temporal evolution of the tropical ozone response to the solar rotational cycle are examined by calculating the cross-correlations between filtered F205 and ozone in observations and model results. Results are shown in Fig. 6. For 1991-94, the observations exhibit a cross-correlation peak at 0.28 on the 4.6 hPa level with no time lag (Fig. 6a). This maximum value is close to the maximum of 0.35 found by Hood and Zhou (1998) on the same pressure level. Furthermore, the overall variation of the time lag with altitude shown in Fig. 6 is similar to that found in previous studies (Hood, 1986;

Brasseur et al., 1987; Brasseur, 1993; Hood and Zhou, 1998) with a negative lag above 3-4 hPa (ozone "leading" the solar flux) and a positive lag below (ozone lagging the solar flux). As mentioned in the introduction, the negative lag in the upper stratosphere results of the influence of the temperature feedback on the ozone response through the temperature dependent chemical reactions. For 2004-07, the cross-correlation pattern (Fig. 6d) is more distorted and weaker than for 1991-94 (Fig. 6a). The cross-correlation maximum (0.2) is smaller than for 1991-94 and is found at 10 hPa with a time lag of +5 days (ozone

lagging solar flux).

Although the cross-correlation fields for the CTM and CCM simulations appear smoother and with larger statistically significant (shaded) areas than for the MLS data, most of the general features present in the MLS cross-section fields appear consistently reproduced by the simulations in the two model configurations. Marked differences between the CTM and the observations are found in 1991-94 though. The high correlation area (with a maximum of 0.4 at 7 hPa and a positive time lag

of 3 days) expanding throughout the middle stratosphere (between 30 and 10 hPa) in the CTM (Fig. 6b) is not found in observations (Fig. 6a). Overall, the main area of significant correlation appears also lifted upward in the observations (Fig. 6a) compared to the CTM (Fig. 6b). The fact that the correlation signal in the middle and lower stratosphere (below 10 hPa) is found in the CTM but not in the observations may partly arise from the large noise present in the UARS MLS ozone dataset at these altitudes (not shown). In contrast, the results for the period 2004-07 reveal a particular good agreement throughout

stratosphere between the observations (Fig. 6d) and the CTM (Fig. 6e), where the maximum is found at the same altitude (10 hPa), time lag (+4 days) and with the same amplitude (0.2). CCM results show a maximum of correlation also at 10 hPa and at the same time lag but with a higher value (0.3). In addition to the area of statistical significance which increases when examining CCM results, we notice a strong reduction of the discrepancies in the response between both periods. This suggests that averaging over the five ensemble members allows to reduce the effect of the non-solar random variability in the signal

estimation and hence to identify more robustly the solar signal. Nevertheless, for 2004-07, we note a weaker correlation and a reduced downward propagation of its extension which is likely due to a weaker rotational UV forcing compared to 1991-94 (Figs. 1 and 2).

Above 3 hPa (~40 km), CCM cross-correlations of both periods (Fig. 6c,f) show a maximum at negative time lag (-2 days). As mentioned in the introduction, this negative time lag can be induced by temperature feedback on ozone and by increasing

hydrogen radical HOx from enhanced solar irradiance which contributes to ozone destruction. While our model configuration allows to fully account for the HOx effect, the solar-induced temperature response is limited since the direct radiative heating effect is not included in the model. The temperature response to the 27-days cycle is thus solely controlled by the ozone concentration change (caused by photolysis changes) and not from the direct heating effect driven by solar irradiance change.

Although a temperature signal is found (not shown), it is small, hence reducing the likelihood for the solar-induced temperature feedback to be prominent in our experiments. It is interesting to notice that the upper stratosphere negative lags in our experiments compare very well with those found in CCM experiments of Sukhodolov et al. (2017) (see their Fig. 3) despite the fact that their model (SOCOL) also includes the direct radiative heating effect. At first glance, this good agreement with our model results may suggest that neglecting the direct effect on heating rates generated by UV variations has a limited effect

on the ozone response, at least at 27-day timescales. However, this conclusion cannot be drawn because the two models have different photolysis, chemistry and radiation schemes. In particular, it has been shown recently that the photolysis rates calculated by LMDZ-Reprobus and SOCOL can differ substantially (Sukhodolov et al., 2016). The good correspondence between the two sets of model results may thus be fortuitous. For instance, the difference in the photochemical response between SOCOL and LMDz-Reprobus could be compensated by the direct heating rate effect included in SOCOL. Also, a

better evaluation of the impact of the direct radiative heating effect requires to perform LMDz-Reprobus experiments, with an increased spectral resolution of the radiative scheme, which account for daily fluctuations of the SSI.

In addition to correlation analysis, ozone response to solar UV flux changes can also be measured in terms of sensitivity, i.e. percentage change in ozone per 1% change in solar UV. Considering ozone sensitivity instead of ozone absolute change allows in principle to analyse an ozone signal that does not depend on the magnitude of the solar rotational forcing, assuming implicitly

that the relationship between the solar forcing index (F205) and the ozone response is linear. We derive the ozone sensitivity on different pressure levels by linear regression of the filtered ozone time series on one independent variable, F205. In previous studies, ozone sensitivity profiles were either calculated at optimum lags where the correlation coefficient maximizes (e.g. Hood and Zhou, 1998) or at zero lag (e.g. William et al., 2001; Austin et al., 2007). Both alternatives were tried but given the limited effect on the results and conclusions, we elected to show only ozone sensitivity profiles using a common time frame,

hence at zero lag. Results are shown on Fig. 7.

For the 1991-94 period, the observational (UARS MLS) sensitivity peaks at 0.4 (0.4% of ozone change for 1% change in F205) near 4-5 hPa (35 km), consistent with the results of Hood and Zhou (1998) (Fig. 7a). For the 2004-07 period, the shape of the observational (Aura MLS) sensitivity profile is distorted and the sensitivity peaks at only 0.2 around 5 hPa (Fig. 7d); it is consistent with a peak value of 0.15 derived at the same level shown in Dikty et al. (2010) for a similar period (2006-07) but

with a different instrument (ENVISAT SCIAMACHY). In the middle stratosphere, the sensitivity profile calculated from the CTM results for the period 1991-94 (Fig. 7b) is consistent with the MLS sensitivity profile (Fig. 7a); the CTM sensitivity profile peaks at 4-5 hPa with a value slightly lower (0.3) than that derived from the MLS observations. Discrepancies between CTM and observational sensitivities are more pronounced in the upper stratosphere. In the CTM, above the peak, the sensitivity suddenly drops around 3 hPa to values close to 0 (Fig. 7b), while in the observation the sensitivity gradually decreases from

3-4 hPa to the stratopause region (around 1hPa) (Fig. 7a). Below 10 hPa, we also note that the uncertainties of the sensitivity profile estimates are larger in the observations than in the CTM. This is consistent with the absence of solar-ozone correlation signal at these altitudes in the observations (Fig.6a) and, inversely, the clear solar-ozone correlation signal in the CTM (Fig. 6b). For 2004-07, the CTM sensitivity profile appears to be highly consistent with observations throughout the stratosphere,

in accordance with the previous coherence and correlation analyses (Figs. 4 and 6).

We now analyse the CCM ensemble results. The ensemble mean ozone sensitivity profiles (Figs. 7c and f) markedly differ with ozone sensitivity profiles derived from observations (Figs. 7a and d) and CTM (Figs. 7b and e) at the corresponding periods. These differences are particularly pronounced in the upper stratosphere (above ~5 hPa). On the other hand, despite the two different periods, the ensemble mean ozone sensitivity profiles show very similar features with positive sensitivity

from 15 hPa to the stratopause and a maximum sensitivity of 0.4 at ~3 hPa (Figs. 7c and f). This maximum tropical sensitivity value and its altitude level is in good agreement with previous CCM estimates (e.g. Rozanov et al., 2006; Austin et al., 2007; Gruzdev et al., 2009; Kubin et al., 2011). The CCM ozone sensitivity analysis has also been repeated for the period 2003-2005 (not shown) to be directly comparable with the CCM results of Sukhodolov et al. (2017): like for the correlation analysis (Fig. 6), we found very similar ozone sensitivity profiles. The ensemble spreads (i.e. 2σ standard deviation calculated over the five

CCM simulations for each 3year period, dashed line) are of the same order for both periods (Figs. 7c and f). They are also very large, indicating important variations from one ensemble member to another, which are most likely due to differences in dynamical variability. Similar conclusions have been reached in previous CCM studies (e.g. Rozanov et al., 2006; Austin et al., 2007). This may partly explain the strong differences in ozone sensitivity found between the two periods in the observations and the CTM simulation. In a sense, each 3year observed period can be viewed as a single realization of an ensemble.

As mentioned in Section 2.2, the results based on UARS MLS measurements may be affected by the imbalance between night and daytime sampling due to the ozone diurnal cycle becoming significant in the upper stratosphere. To test the influence of the ozone diurnal cycle, we repeated all the analysis performed in this section by mimicking an irregular sampling over the period covered by Aura MLS (i.e. 2004-2007). Each day, ~700 ozone vertical profiles of the Aura MLS instrument are evenly retrieved in the tropics [20S-20N] at two fixed local times: one at night (~0142 LST) and one during daytime (~1342 LST).

We initially build the ozone time series using daytime measurements only (1095 days in total). Among these 1095 days, we selected N days randomly where daytime measurements were replaced by night time measurements. We then repeated the spectral, correlation and regression analysis. The procedure was performed for various values of N, from N=100 (i.e. 91% of daytime measurements) to N=1000 (i.e. 9% of daytime measurements). The results (not shown) revealed almost no dependence to N, suggesting that the diurnal cycle has a small effect on the ozone solar rotational signal.

Overall, our results demonstrate that the LMDz-REPROBUS model produces an ozone response to the solar rotational cycle that is consistent with observations, especially when the dynamical variability is accounted for in the analysis. The results of our ensemble of transient CCM simulations further support the importance of atmospheric internal variability in modulating or masking the solar signal in ozone at solar rotational time scales. In the following, we exploit the ensemble simulation to examine thoroughly the temporal variability of the ozone sensitivity to the rotational cycle.

## 4 Temporal variability of the ozone response sensitivity

### 4.1 Does ozone sensitivity to the rotational cycle depend on the amplitude of the solar fluctuations?

Results from CCM studies of Gruzdev et al. (2009) and Kubin et al. (2011) suggested that ozone sensitivity seems to decrease with increasing amplitude of the rotational cycle. The amplitude of the rotational cycle depends on the inhomogeneous brightness structure of the solar disc (i.e. distribution of sunspots and faculae). Given that the amount of sunspots and faculae increases with increasing solar activity, inhomogeneity in the brightness is likely to increase during solar maximum phases. One may thus expect minimum and maximum sensitivity during 11year solar maximum and minimum phases, respectively. Next, we test this hypothesis by dividing 15 years (1991-2005) of the CCM simulations into five 3year windows corresponding to the four different phases of the 11year solar cycle (*i.e.* maximum, minimum, descending, ascending phases). These time windows are highlighted with different colours in the insert panel of Fig. 8a. Figures 8b-f show, for each 3year time window, the ensemble mean sensitivity profiles and the associated $2\sigma$ ensemble spread. The ensemble mean for a specific 3year window is calculated by first computing the ozone sensitivity over this specific 3year interval for each of the five ensemble members and then averaging theses five sensitivities; we define the ensemble spread as the ensemble $2\sigma$ standard deviation. Note that, in total, 15 years of model data are taken into account for the calculation of the ensemble mean sensitivity.

Whatever the solar cycle phase considered (Fig. 8a), all the mean sensitivity profiles have similar shapes with a maximum at around 3 hPa, consistent with observed and modelled sensitivity profiles during solar declining phase (Fig. 7). The most pronounced difference is the maximum sensitivity which varies between 0.3 (green) and 0.5 (red). Overall, the ensemble mean sensitivity profiles appear to vary little from a 3year window to another. Thus, the model ensemble mean ozone sensitivity seems to be rather independent of the level of solar activity (Fig. 8a), at least when 15 years of model data are considered in total. In comparison, the model ensemble spread is clearly more sensitive to the 11year solar cycle phase than the ensemble mean. The ensemble spread is found to be generally smaller during high solar activity periods. It is not surprising. The estimation of the ozone sensitivity is expected to be less affected by the noise and more robust when the solar rotational fluctuations are stronger: the amplitude of the ozone response is much greater, improving the signal-to-noise ratio. We also notice that the ensemble spread is smaller during the maximum phase of cycle 22 (black) than that of cycle 23 (green). It is consistent with the results of Fioletov (2009) observational study that also shows a stronger rotational periodicity in the upper stratosphere tropical ozone during the maximum phase of the solar cycle 22 than the maximum phase of the cycle 23.

Although the rotational cycle amplitude varies with the phase of the 11year solar cycle, the relationship is not systematic as revealed by the wavelet analysis of Fig. 2. In the following, the ensemble mean ozone sensitivity and its spread are examined as a function of the amplitude of the solar rotational cycle fluctuations using sliding time windows. The analysis focuses on the 3 hPa level where the maximum sensitivity is found (Fig. 8). Figure 9 compares the temporal evolution (from 01/01/1991 to 31/12/2005) of the variance of the filtered F205 time series (Fig. 9b) with the ensemble mean (Fig. 9c) and variance (Fig. 9d) of the ozone sensitivity derived from the five CCM simulations. Each point of the time series is obtained by first calculating the ozone sensitivity for each ensemble member over a 1year time window and then computing the ensemble mean and its

variance over the five simulations. The time window is then shifted by 1 month and the same procedure is repeated. This gives a total of 168 1year time slices (14 years x 12 months).

The mean ozone sensitivity time series (Fig. 9c) on 1year time window strongly fluctuates from 0 to 0.6 around an average value of ~0.4, consistent with the value of the ensemble mean sensitivity profiles at 3 hPa (Fig. 8). These fluctuations increase during the minimum phase of the solar cycle in 1995-1998, indicating a larger uncertainty in the estimation of ozone sensitivity during low solar activity periods. This is further supported by the apparent inverse relationship which is found between the F205 index variance (Fig. 9b) and the ozone sensitivity variance (Fig. 9d). Hence, the accuracy of the ozone sensitivity estimate to solar rotational cycle is degraded when solar rotational fluctuations are small, and reciprocally. Finally, note that the low-frequency (i.e. decadal scales) variability of the ensemble mean ozone sensitivity (Fig. 9c) may also suggest an inverse relationship with the F205 absolute value (Fig. 9a) and its variance (Fig. 9b). In the following, we investigate further the relationships suggested here which link the solar rotational variability to the ensemble mean and spread of ozone sensitivity.

Figure 10 shows the regression analysis of the ensemble mean (Fig. 10a) and spread (Fig. 10b) of ozone sensitivity (i.e. dependent variables) on the solar rotational variance (i.e. explanatory variable). We assess the statistical significance of the regression slope using a block bootstrapping technique to account for the autocorrelation in the residuals that can lead to an underestimation of the standard error (Mudelsee, 2014). The bootstrap procedure is carried out as follows. The original residuals are first obtained by subtracting the original fitted model (i.e. derived from the linear regression) to the dependent variable. The original residual time series is then segregated into moving blocks of length $L$ (see e.g. schematic p74 in Mudelsee (2014)) that are randomly resampled to reconstruct a synthetic residual time series of the same size as the original one. Adding this synthetic residual time series to the original fitted model allows creating a new synthetic time series (so-called bootstrap sample) to which the linear regression is applied to derive a synthetic slope value. For each value of $L$, this procedure is repeated 10,000 times in order to construct a distribution of synthetic slopes (Poulain et al., 2016). Finally, we estimate, from this distribution, the likelihood (*p-value*) for the slope to be greater than - or equal to - 0 (i.e. null hypothesis). Note that since $L$ is not known a priori, the calculation is repeated for $L$=1, 2, 3, ...., 10,…, 20, etc. and the largest *p-value* is retained.

Figure 10a reveals no significant negative trend between the mean ozone sensitivity and the F205 variance. Although the linear regression hints at increasing mean ozone sensitivity for decreasing F205 variance, the likelihood for the slope to be positive or equal to zero cannot be excluded statistically ($p > 0.10$). In addition, a non-significant correlation coefficient of -0.19 between the mean ozone sensitivity and the F205 variance is found. This is not the case for the spread of ozone sensitivity, which significantly increases with decreasing high-frequency (short-term) F205 variability (Fig. 10b). This trend further intensifies for the lowest F205 variance values (black and purple dots), corresponding to the phase of the solar cycle with the lowest activity (see insert panel on Fig. 10b). This quantitative analysis hence confirms that the accuracy of the ozone sensitivity estimation increases when the F205 fluctuations are large. We similarly tested the dependence of the mean ozone sensitivity and its spread to the absolute value of F205 (shown in Fig. 9a), an indicator of solar activity. Results are not shown here for brevity. Although we obtain results consistent with those based on the F205 variance (which is expected given the close connection between solar cycle activity and solar rotational fluctuations), the statistical significance is found to be less

pronounced, suggesting a closer link with the amplitude of the fluctuations of the rotational solar cycle rather than the absolute values of F205.

## 4.2 Influence of the size of the time window analysis

Finally, the robustness of the estimated ozone sensitivity is examined with respect to the size of the time window. The procedure is as follows. For each ensemble simulation (of maximum size $t_{max}$=15 years), a time window of a given size, say $\Delta t$, ($\Delta t$ is comprised between 1 and 15 years) sliding by a 1year step is used to resample the ozone 15year time series and create $n_{ensemble, windows}(\Delta t)$ $(= t_{max}-\Delta t+1)$ shorter time series of size $\Delta t$. Given that the ensemble contains five simulations, the total number of samples for a given $\Delta t$ is thus $n_{windows}$=5 x $n_{ensemble, windows}(\Delta t)$ (i.e. 75, 45, 5 samples for 1, 7, 15year time windows, respectively). For each time window size, the ozone sensitivity to F205 is estimated per individual sample. Finally, the mean ozone sensitivity and its spread are derived by calculating the average and the standard deviation over all samples.

Figure 11a shows the ozone sensitivity profiles when a 1year time window is considered. In agreement with the previous ensemble mean ozone sensitivity profiles calculated for 3year time windows and at different solar cycle phases (Figs 7 and 8), a maximum mean sensitivity of 0.4 is found near 3 hPa. The ozone sensitivity spread (dashed envelop) is larger though and even expands towards negative values, demonstrating that a 1year window is not at all long enough to estimate robustly the ozone sensitivity. Figure 11b focuses on the 3 hPa pressure level, where the sensitivity peaks, and reveals that, as expected, the longer the time window is, the smaller the spread is. Figure 11c shows the coefficient of variation of the ozone sensitivity (1σ standard deviation normalized by the mean and expressed in percent) as a function of the size of the time window. It is found that a minimum time window size of 3 years or 10 years is required for the standard deviation to drop under 50% or 20%, respectively, of the mean sensitivity (i.e. ~0.4). These uncertainty ranges also strongly depend on the amplitude of the solar rotational variations and hence the phase of the 11year solar cycle; we find that during solar maximum of cycle 23, minimum of cycle 22, a minimum time window size of 2, 5 years, respectively, is required for the standard deviation to drop under 50%. To obtain a standard deviation lower than 20%, we however found that randomly choosing a 10year time window length performs better than restricting the analysis to short but solar maximum period only (i.e. solar 23). These results suggest that long time series are preferable to estimate accurately the ozone sensitivity to solar rotational fluctuations in observations. It is very likely that some, if not most, of the discrepancies between estimates of the ozone sensitivity found in previous studies originate from differences in the periods and lengths of the considered time windows.

## 5 Summary and concluding remarks

In this paper, we examined the tropical stratosphere ozone response to the solar rotational cycle in satellite observations and simulations of the chemistry-climate model LMDz-Reprobus. We first focused our analysis on the case study of two 3year periods associated with the declining phases of solar cycles 22 and 23. The solar rotational fluctuations are stronger during the first period than the second period. We found that, although the solar rotational signature in the UV forcing is reasonably well

marked during both periods, the amplitude of ozone variations at the corresponding timescales (i.e. ~27 days), in observational records and individual model realizations, does not differ from the noise. Nonetheless, UV and ozone fluctuations show a statistical significant coherence in the middle and upper tropical stratosphere (above ~30 km, or 10 hPa) at the solar rotational timescales. These results hence suggest that ozone significantly responds to the solar rotational variations but the signal is partly masked by other sources of ozone variability at these timescales, most likely of dynamical origin. Applying the same spectral analysis to the average of CCM ensemble simulations allows reducing the 'masking' effect by random dynamical variability, so that the rotational signal in ozone can be more easily identified and estimated.

Lag correlations and linear regressions have then been used to characterize the vertical profile of the ozone response to the solar rotational cycle in the observations and the model during the same periods. Although these results are consistent with estimates of previous studies (Hood, 1986; Brasseur et al., 1987; Brasseur, 1993; Hood and Zhou, 1998) and a reasonable agreement is found between the MLS observations and the CTM experiments, significant differences are found between the two periods. This may be attributed to differences in solar UV forcing or in dynamical variability between the two periods. Analysis of the CCM ensemble simulations suggest that the differences mostly originate from the dynamical variability. The large spread in the ensemble mean sensitivity profile calculated for 3year intervals reflects the 'masking' effect of non-solar dynamical variability in the estimation of the solar rotational signal in ozone and may certainly explain some inconsistencies found in previous studies.

In our CCM experimental design, the direct radiative effect of UV on heating rates has been neglected leading to an underestimated temperature response to the 27day cycle. As a consequence, this may affect the ozone response significantly by reducing the temperature feedback on chemical reaction rates, notably ozone destruction through the Chapman cycle. Recently, Sukhodolov et al. (2016) examined the separate effects of heating rates and photolysis rates in solar-driven ozone changes using a 1D radiative-convective-photochemical model and different SSI datasets. Using the NRLSSI solar forcing dataset, they showed that, over the course of the 11-year solar cycle, the direct heating rate anomaly leads to a decrease in ozone of 1% in the middle and upper stratosphere (above 30 hPa) while the photolysis induces an ozone increase of 2 to 4%. Since, the direct radiative effect of UV on heating rates is neglected in our CCM experiments, the ozone response to solar variability may hence be overestimated. Nevertheless, a comparison of the ozone response in our analysis with results from previous independent CCM studies (Rozanov et al., 2006 ; Sukhodolov et al., 2017) revealed a very good correspondence, despite the fact that their experimental design included the direct radiative heating effect. This comparison must be considered with caution as Sukhodolov et al. (2016) found substantial differences in calculated photolysis rates between LMDz-Reprobus and SOCOL photolysis codes. Therefore, accounting for the direct heating rate effect in SOCOL may compensate differences between the two models in ozone response controlled by photochemical processes only. In addition, the results of Sukhodolov et al. (2016) are based on 1-D model calculations and may also change when accounting for dynamical variability (i.e. using 3-D CCM), particularly at 27day time scales where the atmospheric internal variability largely dominates stratospheric temperature variability (Sukhodolov et al., 2017). To quantify the impact of neglecting solar-induced temperature feedback on our results, the spectral resolution of the LMDz-Reprobus radiative scheme should also be increased and new experiments

including the direct radiative effect of UV on heating rate should be performed. We further notice that these improvements are necessary to simulate the "top-down" mechanism which is based on dynamical consequences of the upper stratospheric thermal response."

Next, we take advantage of the ensemble of five CCM simulations to test whether the ozone sensitivity depends on the phase of the 11year solar cycle. Considering an ensemble of simulations allows in particular to reduce the masking effect induced by the dynamical random variability. Our results suggest that the level of solar activity does not have an impact on the expected value (i.e. ensemble mean) of the ozone sensitivity. However, the ensemble spread decreases during high solar activity periods, making the ozone sensitivity retrieval easier and more robust, e.g., during the maximum phase of the 11year solar cycle.

The ensemble mean ozone sensitivity and its spread have been additionally examined as a function of the amplitude of (i) the solar rotational cycle fluctuations (shown) and (ii) the phase of the 11year solar cycle (not shown). Here again, no robust dependence of the ensemble mean ozone sensitivity against each of the two variable is found when the results of the five 15year simulations are averaged. Although the results hint at a slightly negative trend, i.e. increasing ensemble mean ozone sensitivity for decreasing rotational fluctuations (or 11year solar cycle activity), neither the slopes nor the correlation coefficients are statistically significant. Hence, our results could not confirm previous findings of Gruzdev et al. (2009) or Kubin et al. (2011) who, using model experiments, suggested an increased ozone sensitivity with decreasing solar rotational fluctuations. Nevertheless, it must be noted that the conclusions of Gruzdev et al. were reached by carrying out experiments with a solar rotational forcing that had an amplitude 3 times larger than a realistic one. Further model experiments, considering for instance longer simulations and/or stronger forcing, would help to address this issue more thoroughly.

In contrast with the ensemble mean ozone sensitivity, as expected, the ensemble spread ozone sensitivity shows a clear increase with decreasing solar rotational cycle fluctuations. The negative trend further intensifies during the period with very low solar rotational fluctuations, corresponding here to the period of minimum solar activity between the end of the solar cycle 22 and the beginning of the solar cycle 23 (i.e. 1994-1997). These findings are consistent with the results of Fioletov (2009) who showed a noticeable difference in the estimate of the ozone sensitivity profile in 1994-1998 by comparison with other periods. Hence, when the solar rotational fluctuations are small, the 'masking' effect of dynamical variability becomes more prominent and makes the estimate of the ozone sensitivity less accurate.

Finally, we demonstrate that, while the mean ozone sensitivity (e.g. ~0.4 at 3 hPa) is more or less independent of the size of the time window (tested from 1 to 15 years) when the results of the five 15year simulations are analysed and averaged, the accuracy of its estimate improves dramatically with increasing size of the time window. We found that, on average, a minimum time window size of 3 years (corresponding to ~40 solar rotational cycles) is needed for the $2\sigma$ uncertainty to drop below 100%. More concretely, this means that if the ozone sensitivity to solar rotational fluctuations is derived over only three successive years of observations (or of a single model realization), there is a 95% likelihood for the estimate to take any value in the range [0-0.8] at 3 hPa. The error in the sensitivity estimation also strongly depends on the amplitude of the solar rotational fluctuations and is thus linked to the solar activity. For a constant uncertainty threshold, the higher the solar activity is, the

shorter the required time window length is. We finally find that a minimum of 10 years of data is required for the $1\sigma$ uncertainty in the ozone sensitivity estimate to drop under 20%.

Overall, it is likely that the discrepancies in the estimated value of ozone sensitivity found in previous studies originate from differences in the length of time windows that were used for analysis and in the level of solar activity associated with these periods. Both parameters significantly influence the accuracy of solar rotational signal estimates. In this regard, it is likely that similar issues have also affected the accuracy in the estimation of ozone response to the 11year solar signal. The estimation is expected to be even more difficult because observational time series cover a very limited number of 11-year cycles and there are other well-known sources of decadal variability in the atmosphere and climate system. Maycock et al. (2016) recently found very large discrepancies in the estimation of the ozone response the 11year cycle using various satellite datasets which cover different time periods of different length.

## 6 Data availability

UARS MLS and Aura MLS satellite data are publicly available at https://earthdata.nasa.gov/ after registration. LMDz-Reprobus data used in this study are available upon request to the corresponding author (remi.thieblemont@latmos.ipsl.fr).

## 7 Acknowledgments

This project was supported by the European Project StratoClim (7th framework programme, Grant agreement 603557) and the Grant 'SOLSPEC' from the Centre d'Etude Spatiale (CNES). RT was supported by a grant from the LABEX L-IPSL, funded by the French Agence Nationale de la Recherche under the 'Programme d'Investissements d'Avenir'. The authors thank the three anonymous reviewers for their detailed review comments, which improved the manuscript.

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

**Figures**

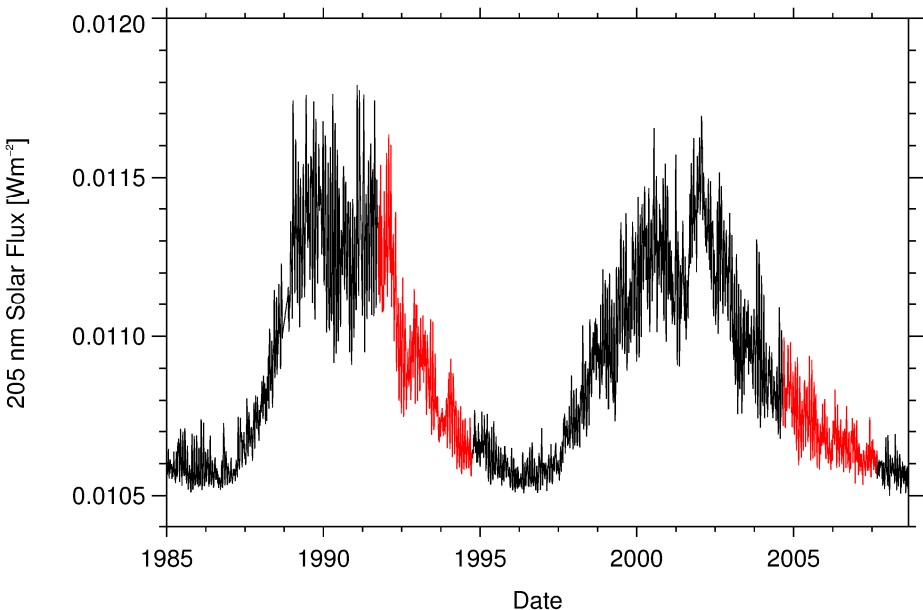

**Figure 1: Temporal evolution of daily F205 from NRLSSI model over solar cycles 22 (1985-1996) and 23 (1996-2008). The two 3year periods considered here (1991-94 and 2004-07) are highlighted in red.**

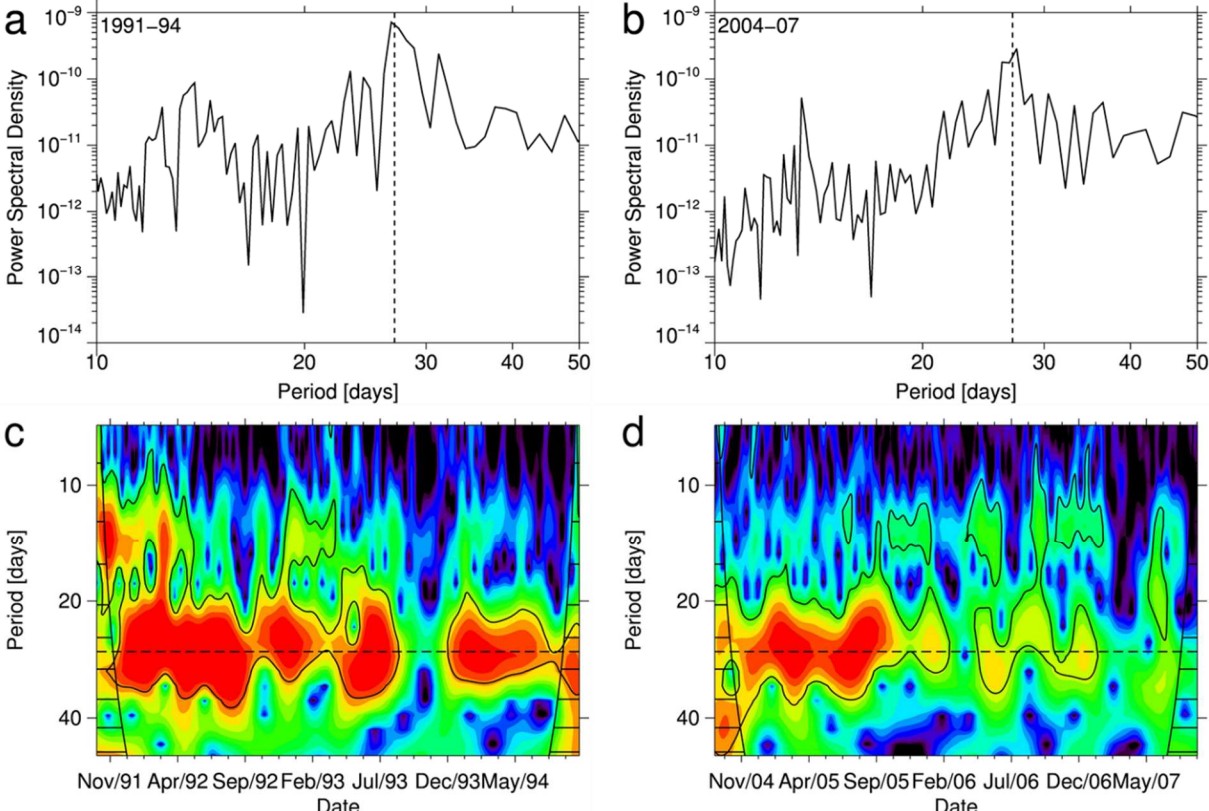

**Figure 2: (Top)** F205 FFT power spectra (from NRLSSI model) for the (a) 1991-94 and (b) 2004-07 period. **(Bottom)** Time-resolved power spectra densities (or scalogram) estimated from continuous wavelet transform (CWT) for the (c) 1991-94 and (d) 2004-07 period. The vertical, horizontal, dashed lines on (a,b), (c,d), indicate the 27day period. The cone of influence, i.e. limit beyond which scalogram should not be interpreted, is marked by horizontal solid stripes. The solid contour lines represent the 95% confidence level.

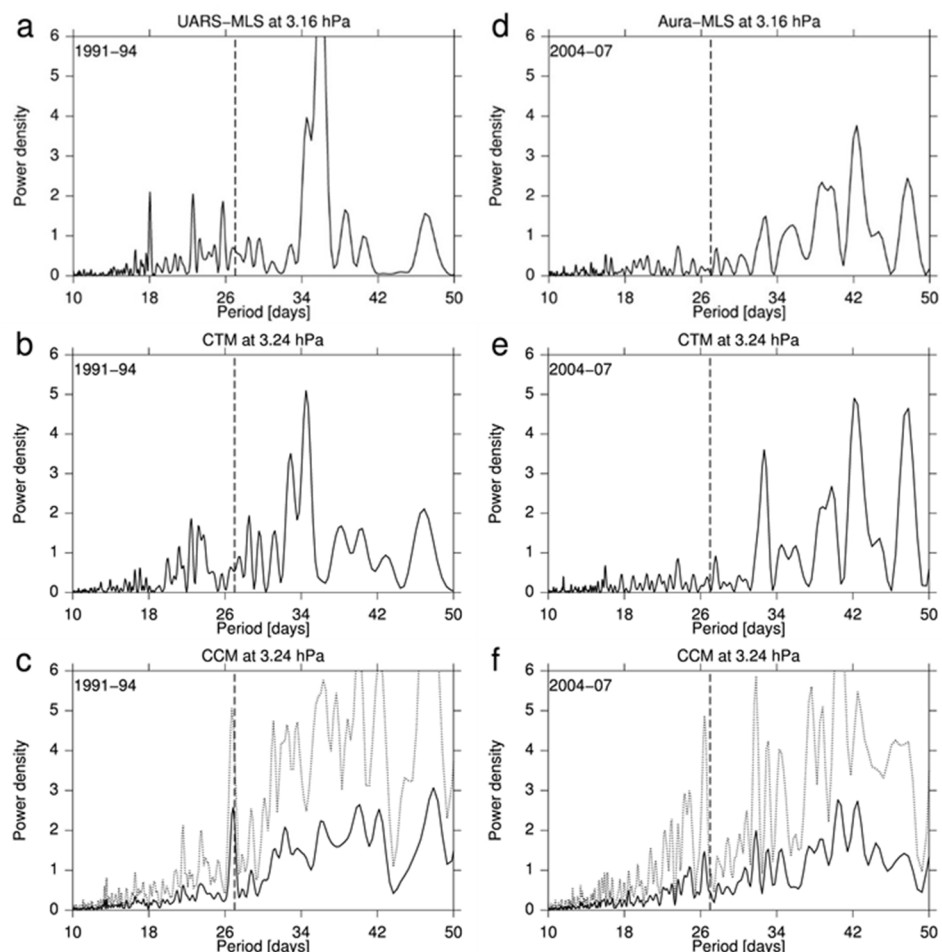

**Figure 3:** Ozone Lomb-Scargle periodograms for the (left) 1991-94 and (right) 2004-07 periods. The top panels represent ozone Lomb-Scargle periodograms from (a) UARS MLS and (d) Aura MLS observations. The middle panels (b and e) represent the ozone Lomb-Scargle periodograms for CTM simulation and the bottom panels (c and f) the average periodogram of the CCM ensemble. The dotted envelop (c and f) indicates the 2σ standard deviation of the ensemble of CCM simulations.

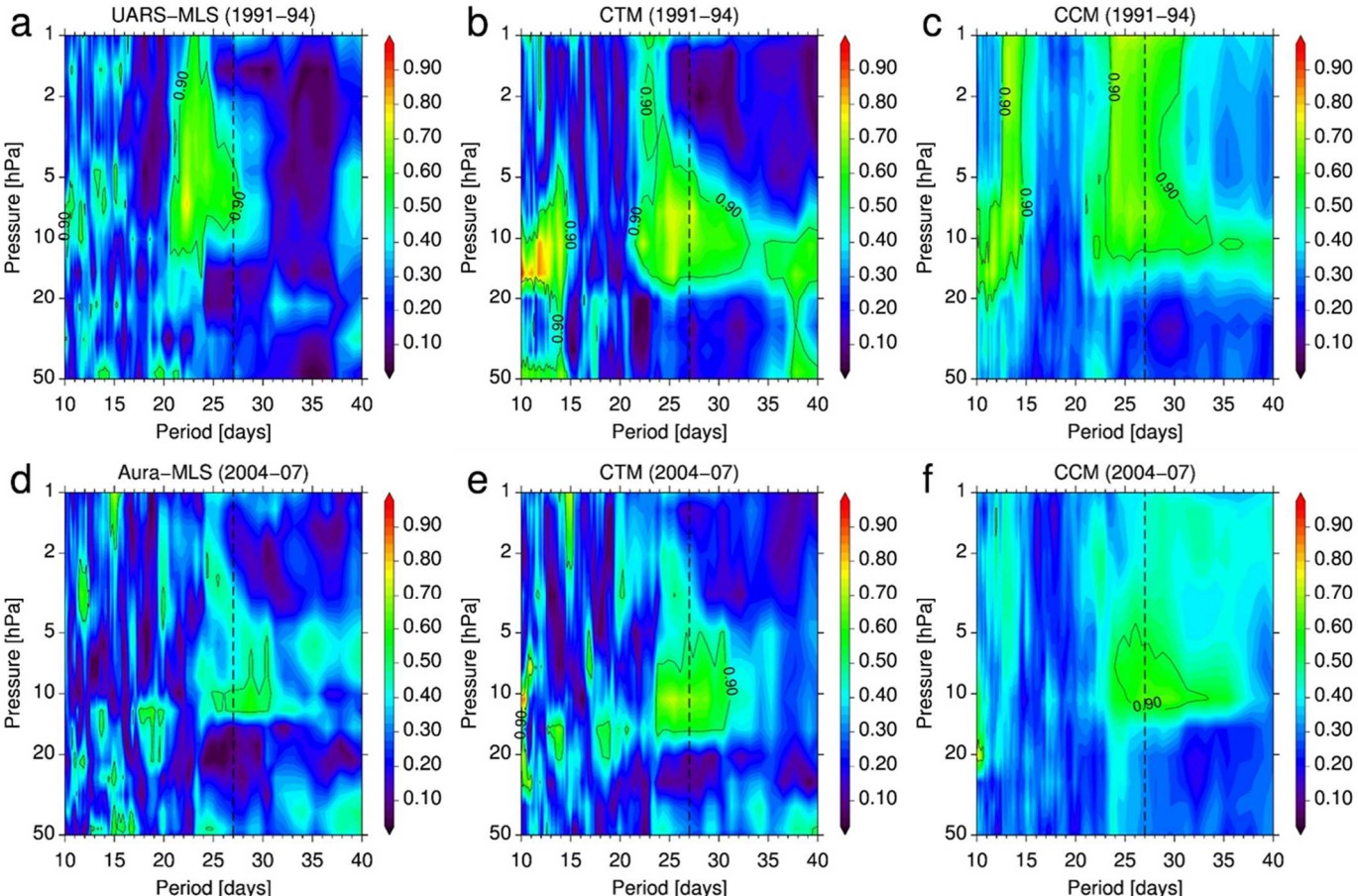

**Figure 4: Mean squared coherence between ozone and F205 as a function of period (days) and pressure level (hPa) for the (top) 1991-94 and (bottom) 2004-07 period and for (a, d) MLS observations, (b,e) CTM and (c,f) CCM simulations. Black contour lines indicate the 90% confidence level and the vertical dashed black lines indicate the 27day period.**

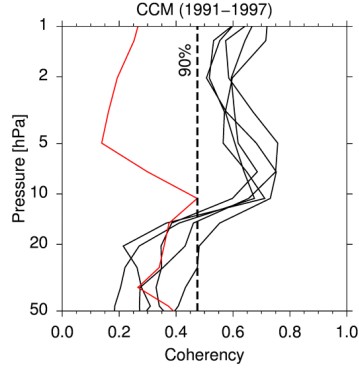

**Figure 5. Vertical profile of the mean squared coherence between ozone and F205 averaged between 22 and 30 day periods and calculated for the time period 1991-1997. The black lines correspond to the results of individual ensemble members (five in total) and the red line to the results of the experiment forced with constant solar forcing. The vertical dashed line indicates the 90% confidence limit.**

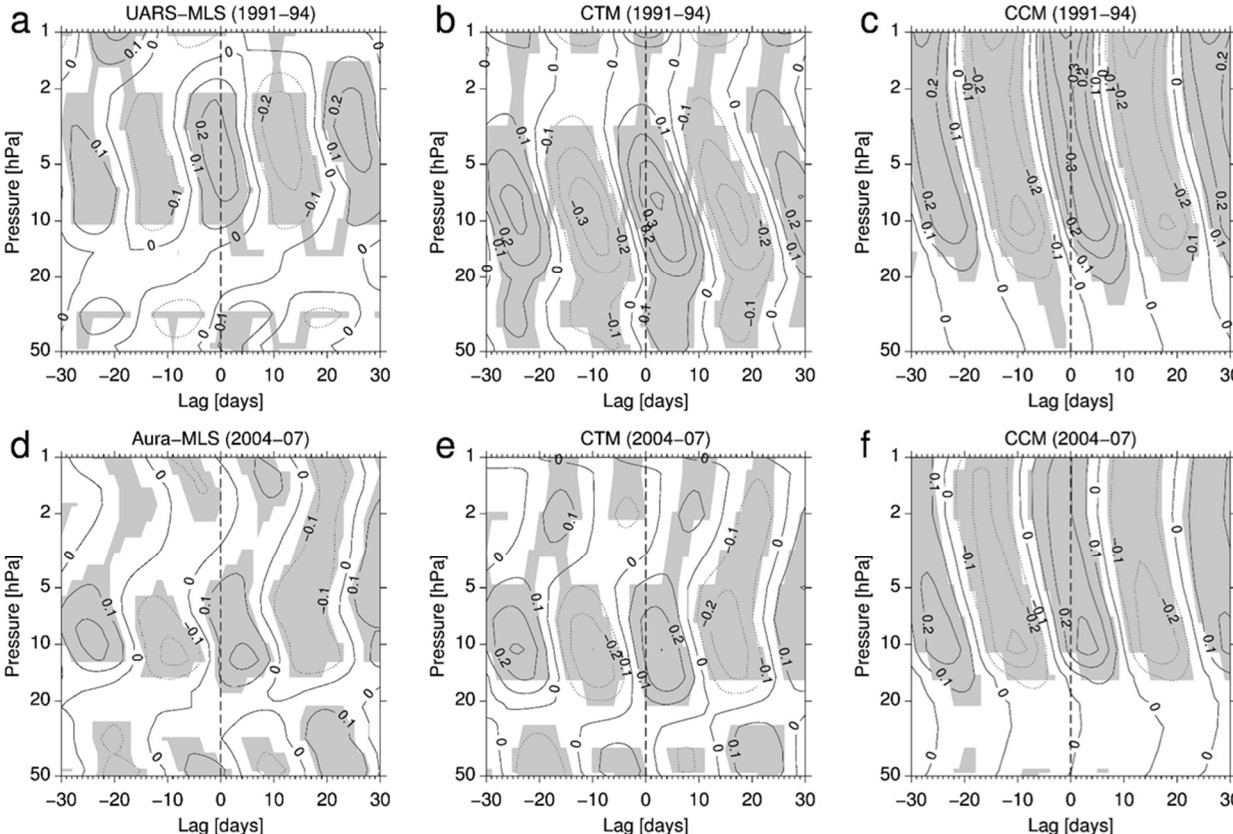

**Figure 6: Cross-correlation between digitally filtered (see main text) ozone and F205 as a function of time lag (in days) and pressure level (hPa) for the (top) 1991-94 and (bottom) 2004-07 periods. (a,d), (b,e) and (c,f) panels show cross-correlation between F205 and MLS observations, CTM and CCM simulations, respectively. Shading represents areas with 95% confidence level.**

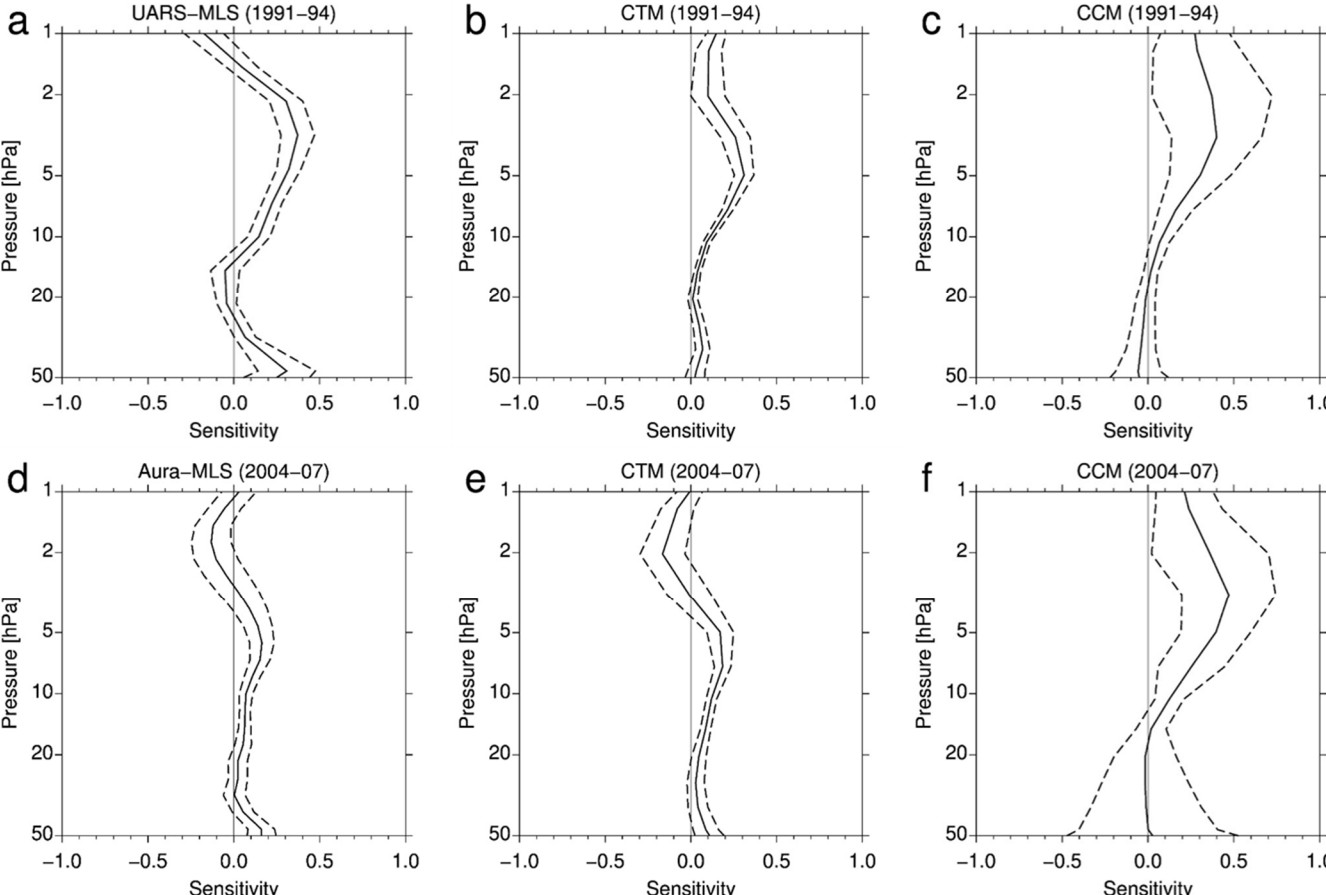

**Figure 7: Vertical profile of ozone sensitivity to F205 (% change in ozone for 1% change in F205) at lag 0 for the (top) 1991-94 and (bottom) 2004-07 periods. Results are shown for (a) UARS MLS, (d) Aura MLS, (b, e) CTM simulations and (c,f) CCM ensemble simulations. (a,b,d,e) The dashed envelop indicates the 2σ standard error of the regression estimates. (c,f) The dashed envelop indicates the 2σ ensemble simulations spread.**

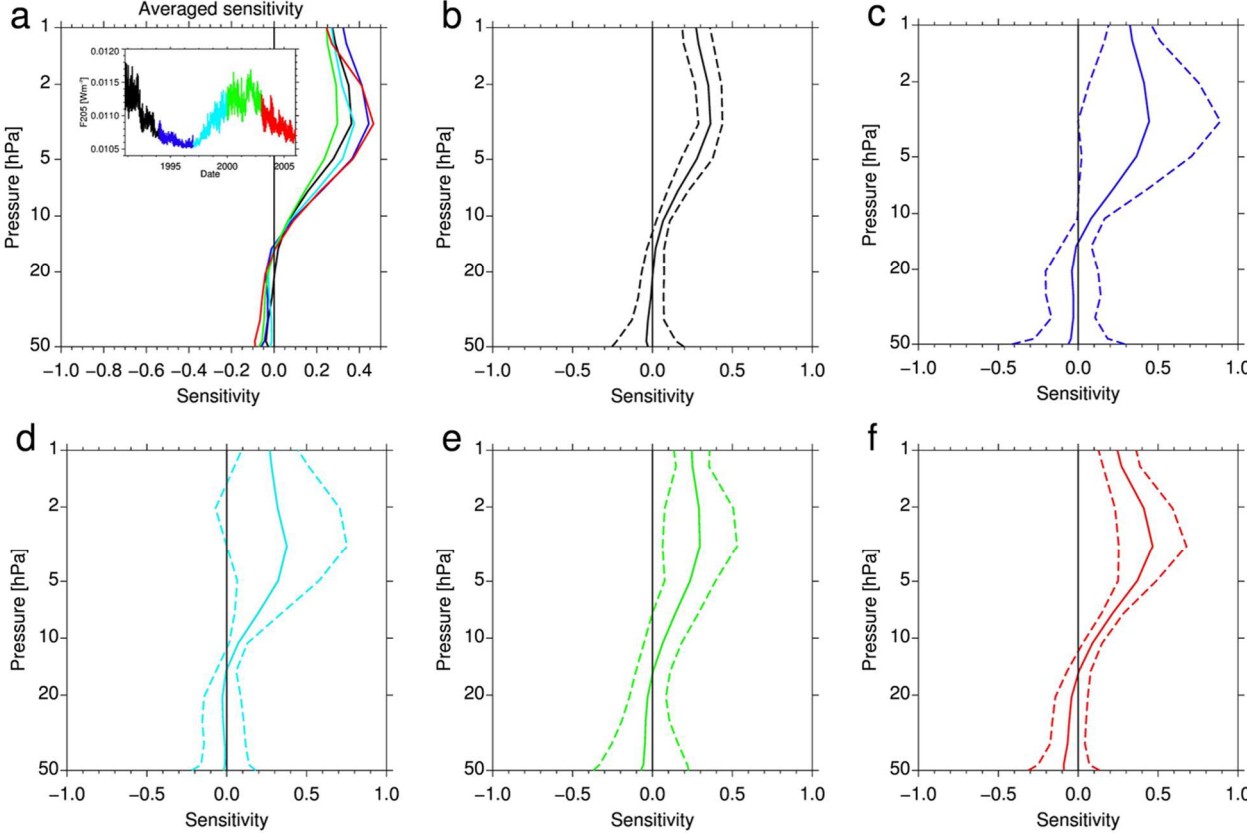

**Figure 8: (a) CCM ensemble ozone sensitivity profile at lag 0 for each of the 3year period. Each period and its corresponding colour is shown in the insert plot (a). CCM ensemble mean ozone sensitivity profile and its 2σ range are shown for each individual 3year periods: (b) 07/1990-06/1993, (c) 07/1993-06/1996, (d) 07/1996-06/1999, (e) 07/1999-06/2002, (f) 07/2002-06/2005.**

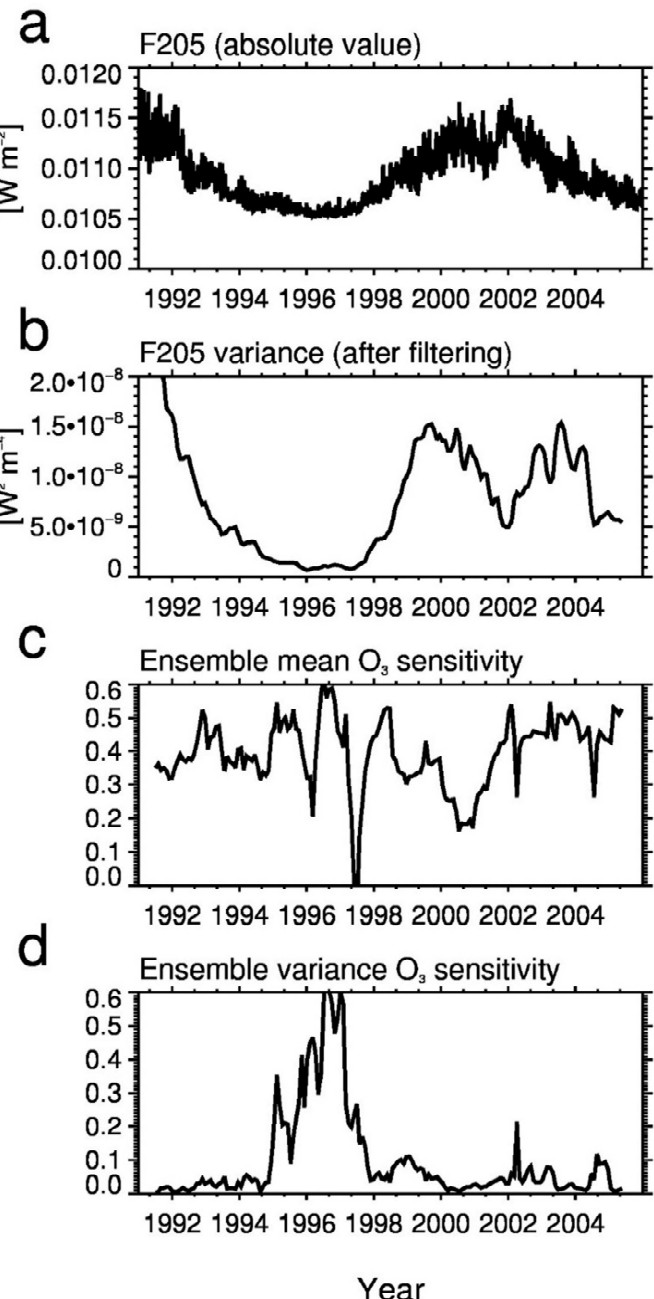

**Figure 9: Digitally filtered (b) F205 variance time series, (c) ensemble mean ozone sensitivity and (d) ozone sensitivity ensemble variance time series at 3hPa computed over a 1year running window. Each window is sliding for one month at each step. (a) The F205 index time series is reproduced on the top panel for clarity.**

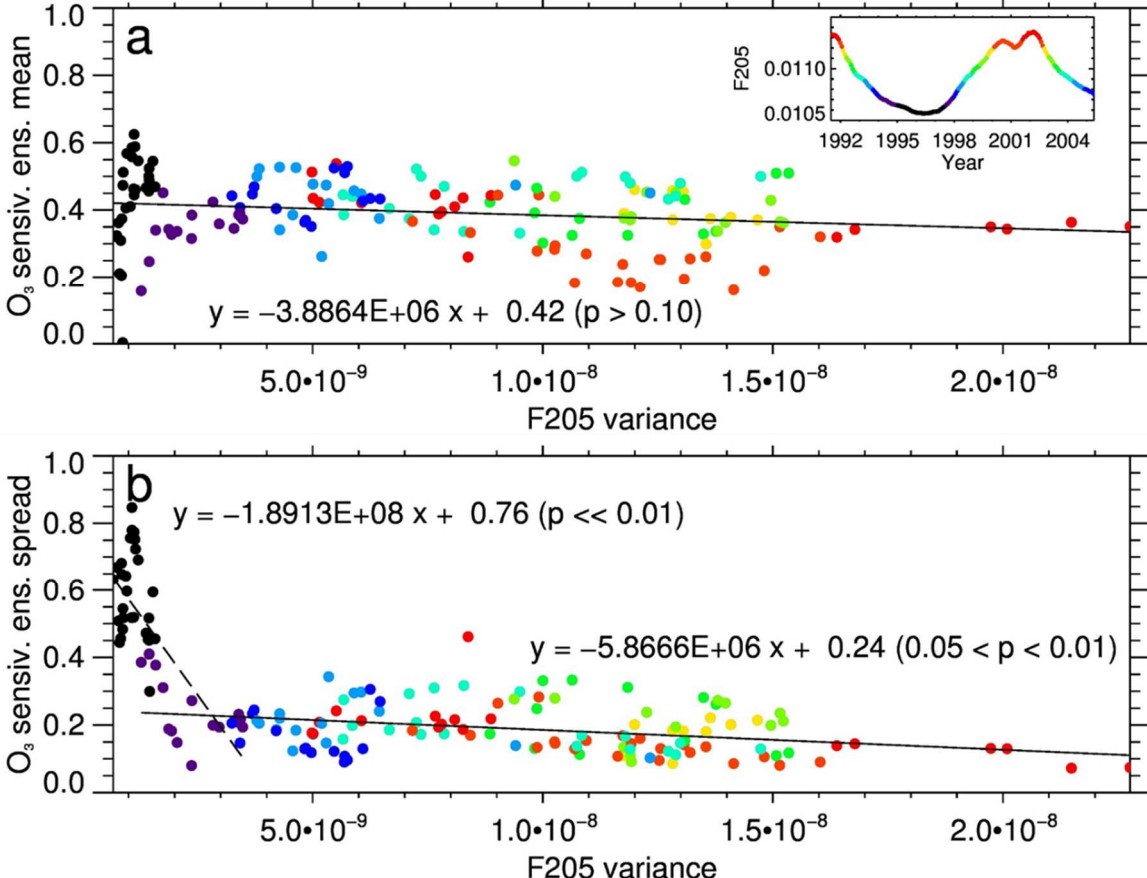

**Figure 10: Scatter plots of the CCM ensemble (a) mean ozone sensitivity and (b) its spread (1σ) versus the F205 variance. Dots are coloured with respect to the value of the F205 flux, shown in the insert plot of panel a. Least square linear regression fits are superimposed (solid and dashed segments) together with their equation and the statistical significance of the slope value (in brackets, see text for details). The correlation coefficients are (a) -0.19 (p>0.10), (b, dashed) -0.76 (p<0.05) and (b, solid) -0.36 (p<0.10).**

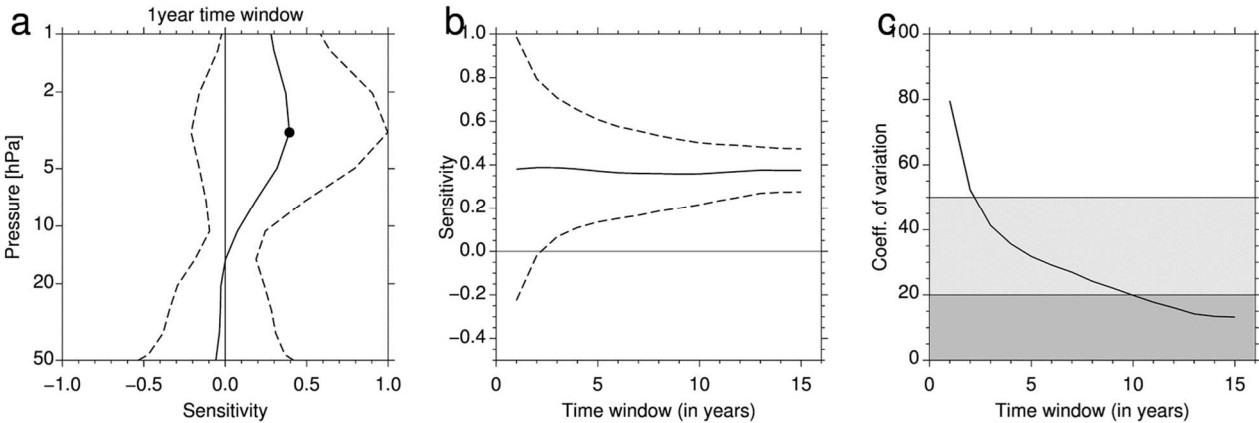

**Figure 11: (a)** CCM mean ozone sensitivity profile over the 1991-2005 period computed for a 1-year time window (see text for details on calculations). **(b)** Mean ozone sensitivity at 3 hPa (dot on (a)) as a function of the size of the time window. The dashed lines on (a) and (b) represent the 2σ spread. **(c)** Coefficient of variation (in %) of the ozone sensitivity as a function of the size of the time window. Intervals with values lower than 50% and 20% are highlighted by the gray shaded areas.