# Peer review of "Sensitivity of the tropical stratospheric ozone response to the solar rotational cycle in observations and chemistry-climate model simulations"

_Atmospheric Chemistry and Physics, 2016_

## Referee Comment (RC1) · Anonymous Referee #2 · 18 Feb 2017

Overall, this is a valuable comparison study of the ozone response to short-term solar UV variations in both observations and a state-of-the-art chemistry climate model. The analysis is detailed and the results offer plausible explanations for differing results obtained in observations covering different time periods. Final publication is certainly expected in ACP. However, I have some important comments that will require some revision.

Main Comments:

(1) In the description of the adopted CCM configuration in section 2.3 (p. 7), the authors say: "We do not take into account the direct effect on heating rates generated by UV variations because previous modelling studies have already shown that the stratospheric ozone response to solar variations is almost entirely driven by the effects of UV changes on the photolysis rates, in particular the photolysis of molecular oxygen (Swartz et al, 2012)." Even on the 11-year time scale when a steady-state approximation is allowed and both photolysis and radiative heating are accounted for, temperature feedback reduces the ozone response in the upper stratosphere at 2 hPa by about 30% compared to that calculated by considering changes in photolysis only (see Figure 2 of Swartz et al.). 30% is still a fairly large fraction and should not be neglected. On the 27-day time scale, it is more important to include radiative effects on temperature and their feedbacks on the ozone response for two reasons. First, on this time scale, the temperature response peaks at a positive phase lag. As reviewed in the Introduction (lines 5 to 14 on p. 3), the lagged temperature response significantly alters (reduces) the ozone response and shifts it to a negative phase lag in the upper stratosphere. Second, as also reviewed there, a dynamical component of the response is produced in the upper stratosphere which feeds back into the temperature response resulting in a larger effect on the ozone response than would be predicted by a 1D radiativephotochemical model. Therefore, please modify section 2.3 to note and discuss these issues and whether the neglect of the modeled temperature response (and its accompanying dynamical response) may lead to errors in the CCM results that would not be present in simulations done in the CTM mode (forced using observed dynamics and temperatures).

(2) Figure 6 compares the vertical profile of the ozone sensitivity to the solar UV (per cent change in ozone for a 1 per cent change in solar UV at 205 nm) as derived from observations for two time periods, from the model using specified temperatures and dynamics (CTM), and from the model in a free-running mode (CCM). While the observational and CTM results agree fairly well, the mean CCM results show a much larger
response in the upper stratosphere than is seen in either the observations or the CTM results. There is apparently no mention of this disagreement in the manuscript. In view of comment (1) above, it seems possible that part or all of the disagreement is due to neglect of the UV-induced temperature response in the CCM, which would modify both the amplitude and phase lag of the modeled ozone response. The sensitivity calculation is apparently at zero lag so it does not take into account the actual phase lag of the ozone response. Therefore, please modify the results and conclusions sections to consider the possibility that the chosen CCM configuration does not accurately simulate the net ozone response in the upper stratosphere (taking into account both the radiatively and the dynamically induced temperature response).

Other Comments:

(3) Introduction, first paragraph, last sentence. "A thorough understanding and accurate quantification of the UV variability effect on the middle stratosphere from which the "top-down" theory stems, are thus necessary." If so, then why is the CCM configuration limited to only the photochemical ozone response? The thermal response and its associated dynamical response are the main components of the top-down mechanism for solar influences on the troposphere.

(4) Section 2.1, line 11. Are you using the NRL SSI version 1 or version 2? It is fine if you are still using version 1 but it should be clarified. Version 2 is available from https://data.noaa.gov/dataset/ noaa-climate-data-record-cdr-of-solar-spectral-irradiance-ssi-nrlssi-version-2

(5) Section 2.2, line 24. Please specify the pressure levels for ozone retrievals for the two MLS instruments. Which versions of the UARS MLS and AURA MLS data sets are being used for this analysis? Please reference more up-to-date descriptions of these data. The Version 5 UARS MLS data set is described by Livesey et al., JGR, v. 108, doi:10.1029/2002JD002273, 2003. Please give URLs where readers who wish to
repeat the analysis can download the data. For example, the UARS MLS data are at https://mls.jpl.nasa.gov/uars/data.php.

(6) Section 2.2, line 31. If only 30% of the measurements are in the daytime, another problem arises, which is the ozone diurnal cycle. This cycle becomes important at roughly 2 hPa and above. Including 70% of measurements at night will therefore have the effect of reducing the estimated ozone response to solar UV variations at 2 hPa and above. This will not affect comparisons with the CTM and CCM provided that the model "measurements" also include both day and night data. Ideally, there should be 70% night and 30% day model data to allow an exact comparison. Please add text to explain this.

(7) Section 2.3, line 24. 39 levels and 70 km lid means a resolution of less than 2 km. This is much better than the MLS vertical resolution, which is about 6 km. One should mention this before making direct comparisons in the following sections.

(8) Figure 1. The units should be  $W/m^2/nm$ .

(9) Section 3.2, Figure 3. The periodogram of the MLS ozone measurements (Figure 3a,d) is done at 3.2 hPa. But, according to Livesey et al. (2003), the UARS MLS measurements were not retrieved at this level, only at 2.2 and 4.6 hPa. So, how are data obtained at 3.2 hPa?

(10) Section 3.2, lines 10-12. Please note that the lack of an obvious solar rotational signal in the MLS data considered here is partly because the measurements were obtained during the declining phases of solar activity using a limb sounding instrument, whose measurements are spatially and temporally sparse. The ozone signal is more easily detectable and repeatable in daily zonal means of nadir-viewing backscattered ultraviolet measurements under solar maximum conditions when solar UV variations are stronger and more coherent. The CTM simulations are also affected by the rela-

**ACPD**
tively weak solar rotational UV variations during the selected time periods.

(11) P. 9, Figure 4. Normally, a cross-spectral analysis should yield phase estimates as well as coherency estimates. There is no mention of phase on p. 9 so it must be assumed that the coherency estimates are at zero lag. But the cross-correlation functions in Figure 5 show that the phase lags are not constant with altitude and are not always zero. They tend to be somewhat negative in the upper stratosphere and become positive in the middle and lower stratosphere. The ozone-UV sensitivities shown in Figure 6 are also presumably at zero lag. This differs from previous observational studies (e.g., Hood and Zhou, 1998) which calculated sensitivities at the so-called optimum lag, i.e., the lag where the correlation maximizes. Please add text to explain that these calculations are being done at zero lag and why this lag is chosen.

(12) P. 10, line 24. Typo: Seizing? Caption to Figure 3: from the runs ensemble?

(13) P. 11, top of page. The CCM results shown in Figure 5c,f are characterized by negative lags near the stratopause. What is the cause of these negative lags? Is it feedback from a temperature response caused only by increased radiative heating associated with the ozone response (holding direct UV heating changes constant)? Or, is it increased photolysis of water vapor in the lower mesosphere and resulting destruction of ozone by odd hydrogen? Or both? Can this be diagnosed?

(14) P. 11, bottom of page. In addition to not mentioning the anomalously large CCM response in the upper stratosphere, there is also no mention here of the likely effect of the ozone diurnal cycle in reducing the ozone response in the upper stratosphere relative to that measured earlier from backscattered ultraviolet instruments, which operated only in the daytime. This difference is emphasized in Hood and Zhou [1998] for example.

(15) Section 4. While it is useful to carry out these analyses, one must question whether

**ACPD**
the CCM in its chosen configuration (no direct solar UV heating changes) is ideal for this purpose. Also, a time window with a length of 10 years includes both solar maximum periods (when 27-day UV variations are strong and numerous) as well as solar minimum periods (when these variations are weak and sparse). Could it therefore be possible that a shorter time window of 3 years centered on a strong solar maximum (e.g., that in 1979-82) could yield more reliable results than a 10-year window which includes mostly non-maximum solar conditions?

(16) Minor corrections: In the abstract, lines 23-24, neither nor should be either or. P. 13, line 10. anti-correlation should be inverse correlation.

(17) P. 15, lines 11,12: "Applying the same spectral analysis to the average of the CCM ensemble simulations allows reducing the 'masking' effect by random dynamical variability, so that the rotational signal in ozone can be more easily identified and estimated." However, the negative aspect of this approach is that the CCM may not perfectly simulate the actual ozone response to short-term UV variations, partly because of the neglect of the direct radiative effet of the UV variations in the model, and their secondary dynamical effects.

(18) P. 15, lines 18-21: "Analysis of the CCM ensemble simulations suggest that the differences mostly originate from the dynamical variability." Usually, internal dynamical variability in a model is larger than in observations so it is not clear that a single model run is equivalent to a single sample of observations (or a single run of the CTM). The large spread in the ensemble mean sensitivity profile could also reflect a less complete simulation of the upper stratospheric dynamical response to short-term solar UV variations.

---

## Referee Comment (RC2) · Anonymous Referee #3 · 18 Feb 2017

The paper uses models and satellite observations to investigate the response of tropical stratospheric ozone to short-term solar UV variations due to the 27-day solar rotation. This is a topic that has been investigated in a number of previous studies but is still not resolved; therefore an updated study will be of interest to journal readers. The stated goals of the present paper are to "(i) assess the influence of the solar cycle phase on the ozone sensitivity to the rotational cycle and (ii) quantify the time window required for a robust estimation of the ozone sensitivity".

While the paper does address the stated topics, the focus is drawn elsewhere by awkward organization and extraneous material. The authors do not make it clear why they include Section 3, which is longer than the section addressing their goals. Section 3 does not contribute to the focus of the paper and, in my view, does not contribute to the understanding of the solar response.

My most serious concern about the paper is in regard to the implicit assumptions in the text. At a number of places, the authors have assumed that a solar response is present even when they do not see a signal of it in their analysis or that the response is larger than what they find from the analysis. The text then describes why and how the signal has been masked. This is dangerous; if you do not find a signal of a response, the first explanation should be that there is no response. Even if processes that might mask a signal are present, it is not appropriate to conclude that this masking is the reason for not finding a signal that is "known" to be present based on prior assumptions. A more appropriate way to say it would be: if there is a response, it is too weak to detect.

Major Comments

1. As indicated above, I did not see the purpose of Section 3. Since you consider two fairly short 3-year periods, the analyses do not have any bearing on the questions raised about variations of the response with timing within the 11-year solar cycle or the dependence on the length of the analysis period.

2. Perhaps the response diagnosed from MLS is intended as validation for your model simulations. In this case, why ignore the 9.5 years of MLS/Aura observations that have been taken since 08/2007? As you later find using model simulations, a 3-year period does not give results that are very robust and so is not convincing as validation.

3. It seems that both satellite data (in Section 3) and model output (throughout) are zonally and latitudinally averaged over each day, including both day and night. There is a mention of local time issues (Section 2.2) but you then decide not to use any local time or day/night information in your analysis. There is evidence that this should be considered: 1) why else would the MLS/UARS ozone variations show a prominent

peak at the yaw period?; 2) it is known that local time variations in the response of ozone in the upper stratosphere to solar variability are not negligible (e.g. Li et al., Earth and Space Science, doi:10.1002/2016EA000199).

4. Although some spread should be expected, if I see a response peaking at 22 days (as you show in Figure 4a), I would automatically assume that it has no relation to the solar rotation. The signal in that particular panel is near zero at 27 days. It seems shaky to interpret it as driven by the solar flux variations. Can you provide more justification for your interpretation?

5. ("additional CCM simulation where the solar flux is kept constant") Since you show that the CCM responses vary considerably between realizations, a single simulation is not a very useful. Alternatives would be to perform additional realizations and/or to perform a similar case (fixed solar flux) with the CTM, particularly for a longer period (>10 years).

6. I am having trouble reconciling Figure 4c, f with Figure 6 and 7. Figure 4 indicates largest signal at $\sim$10 hPa and near zero signal at $\sim$0.3 hPa while Figures 6-7 indicates the opposite. Even though sensitivity (Fig 6-7) is different from absolute response (Fig 4), these do not appear to be consistent. Please explain.

7. Your conclusion (p. 15, l. 18) that "the differences mostly originate from the dynamical variability" is an important one that should be brought out more prominently.

Specific Comments

1. (p. 2) "the life span of a single satellite instrument is generally far less than one solar cycle" This has been true for a few but just as many last for a time span comparable to a solar cycle.

2. (p. 3, l. 24) Why "nonlinear"? Any dynamics will affect ozone.

3. (p. 7) The description of simulated solar flux variation was confusing. You imply that the variations are included in your photolysis lookup table but I could not tell exactly

how. Is the table recalculated every day? Is there a separate table for each value of your solar flux parameter? Please be explicit. Also, the impact of the solar variation on heating rate is not clear. Reading between the lines, I guess what you mean is that the heating will respond to the increased or decreased ozone but that, in the heating part of the calculation, the solar flux is kept constant. Is that what you mean? And one other comment: O2 should also be included on your list of radiatively active gases.

4. (Figure 3) There is a mismatch between the level shown in the figure ($\sim$3 hPa) and the level where you see a response in Figure 4 ($\sim$10 hPa). Perhaps this is related the mismatch between the SAGE/SBUV results, which contributed to the conclusions in the Hood paper you cite to choose the level of maximum response, and other data and models (e.g. see discussion by Dhomse et al., 2016). Also, a better label for the area where you see a signal would be middle stratosphere, not upper.

5. (p. 9, l. 9) "This explains why . . ." This could be the explanation but, as you show later, you are not using enough data to determine a robust signal. It would be safer to say that "This could contribute etc.". Since you cannot see a signal in the observational analysis, it is not appropriate to assume that the response is there but the signal is masked. There may not be a response.

6. (p. 10, l. 28) "The absence of correlation signal in the middle and lower stratosphere in the observations is consistent with the large noise present in the ozone dataset at these altitudes" As in the comment above, this is misleading since it implies that there is an ozone response but it is masked. You have not shown that a response exists.

7. (p. 13, l. 10) "overall anti-correlation" All I see is that there is a period when F205 variance is low and sensitivity variance is high. The curves are otherwise not related and, even in this period, do not follow a similar evolution. Coincidence of one perturbation is not enough to deduce anti-correlation.

Editorial comment
"increasing (decreasing)" and similar construction is grammatically incorrect and very confusing, especially since elsewhere you use parentheses in their legitimate use to define or clarify, e.g. "solar forcing index (F205)".

---

## Referee Comment (RC3) · Anonymous Referee #1 · 20 Feb 2017

The manuscript discusses the responses of the stratospheric ozone response to the solar irradiance variability on the Sun rotation cycle time scale. The authors analyzed the satellite observations by MLS instrument and results obtained with LMDZ model in free running and specified dynamics modes. The subject of the manuscript is appropriate for ACP. Despite many attempts to characterize ozone response to solar irradiance variability several aspects of the problem still remain open. The manuscript is well written and structured, the figures and explanations are clear. The conclusions about the dependence of the ozone response uncertainty on the signal strength and necessary numbers of cycles could be of interest for the scientists working in this area. There are,

however, some flaws which do not allow me to recommend immediate publication.

**Major issues**

1. It looks like the authors intentionally omitted the direct effects of the solar irradiance on the heating rates and temperature. After careful discussion of how important this process for the correct representation of the time lag between ozone and solar UV in the introduction they completely excluded this processes using not correct arguments. The importance of this process is clear even if we put aside the dynamical consequences of the direct radiative heating. I recall the importance of the direct heating was demonstrated in the recent paper by Sukhodolov et al., 2016.

2. It is also not clear why the authors used 3 year period to evaluate ozone response from the observation and model while they show that 3 year time period does not provide statistically robust results (uncertainty only below 50%). This should be somehow explained to the readers. The choice of the number of ensemble runs is also doubtful in the light of the obtained results. If the ensemble run is crucial it would be logical to estimate how many ensemble runs are necessary to reach some kind of convergence. Moreover, the obtained results with free running CCM will be more convincing if the analysis of the CCM runs without solar rotation variability is added.

Minor issues:

1. Page 7, line 1: It is clear since 2004 that two bands scheme cannot be used for the simulation of the atmospheric response to solar irradiance variability. It was confirmed in 2010,2011 and 2014. Several modifications of different complexity are known.

2. Page 7, line 17: I do not understand how climatological temperature can be used in CCM. Please, elaborate. By the way the reference to TUV is confusing. If I am not mistaken in the cited 1999 paper the tropospheric version was described. Maybe it is better to use recent intercomparison of the codes where TUV showed excellent performance relative to reference models. I do not also understand whether or not
daily NRL solar irradiance was used for the photolysis rate calculations.

3. Page 7, line 21-24: I recommend to check carefully this sentence. I do not recall such a bold statement in the cited paper.

4. Page 12, line 9: I think this statement is not completely correct. In general, the magnitude of the rotational cycle depends on non-homogeneity of the dark/bright features distribution, which can be very small for very high level of solar activity.Fig.8 shows for example rather small variances in 2002, while the solar activity level is high.

---

## Author Response (AR1)

The manuscript discusses the responses of the stratospheric ozone response to the solar irradiance variability on the Sun rotation cycle time scale. The authors analyzed the satellite observations by MLS instrument and results obtained with LMDZ model in free running and specified dynamics modes. The subject of the manuscript is appropriate for ACP. Despite many attempts to characterize ozone response to solar irradiance variability several aspects of the problem still remain open. The manuscript is well written and structured, the figures and explanations are clear. The conclusions about the dependence of the ozone response uncertainty on the signal strength and necessary numbers of cycles could be of interest for the scientists working in this area. There are, however, some flaws which do not allow me to recommend immediate publication.

We thank the reviewer for reading the manuscript and providing helpful comments and suggestions. We address the raised issues in turn below.

Our answers to comments and suggestions are written in blue and manuscript changes are written in italic type within double quotes ("*like this*").

Major issues

1. It looks like the authors intentionally omitted the direct effects of the solar irradiance on the heating rates and temperature. After careful discussion of how important this process for the correct representation of the time lag between ozone and solar UV in the introduction they completely excluded this processes using not correct arguments. The importance of this process is clear even if we put aside the dynamical consequences of the direct radiative heating. I recall the importance of the direct heating was demonstrated in the recent paper by Sukhodolov et al., 2016.

We agree that neglecting the direct effect on heating rate must be considered carefully. To test the potential influence of the missing temperature feedback on ozone in our model setup, we performed additional analysis that we compared with recent model results of Sukhodolov et al. [2017].

In their recent study, Sukhodolov et al. [2017] performed an ensemble of 30 simulations for the period 2003-2005 with the SOCOL CCM model. In their experimental setup, they considered both (i) the effects of UV changes on the photolysis rates and (ii) the direct radiative effects on temperature which feeds back on ozone. Hence, their CCM formulation allows accounting for all photochemical and radiative effects. Their large number of simulations and the fact that they simulated the period 2003-2005 using NRLSSI data (like us) allows comparing with our CCM ensemble results in a fair way. Their UV-ozone cross-correlation analysis (see their Figure 3b) reveals a negative lag of 1 to 2 days in the upper stratosphere (above 3 hPa) which is consistent with the one we found in our study (see our Fig. 7c,f). When performing the UV-ozone cross-correlation analysis over the same period (i.e. 2003-2005), we also found a negative lag of 1 to 2 days. We also compared the analysis of the ozone sensitivity to F205 index (see below) over the period 2003-2005. Their ozone sensitivity profiles (left) are shown at optimum lag and ours (right) at optimum lag (solid) and zero lag (dashed). Results from both models compare very well, showing a maximum slightly larger than 0.4 at 3-4 hPa and decreases down to 0.25 near the stratopause. The very good agreement between the two CCMs, despite their different formulation, suggest that omitting the direct radiative heating does not have a strong effect on the ozone response and is therefore an acceptable approximation in our case.

[Figure]

Globally, the manuscript has been revised at several places to discuss the CCM configuration.

- More specifically in section 2.3, we modified the text as follows:

"Thus, the solar rotational cycle forcing is taken into account by using daily photolysis rates calculated by TUV in the photochemistry module of LMDz-Reprobus. *Note however that the direct effect on heating rates generated by UV variations associated with the 27-day rotational cycle is neglected: i.e. daily changes in the spectral irradiance are not considered by the model radiative scheme. As a consequence, part of the thermal and dynamical responses to the 27-day rotational cycle and hence their effect on ozone (through transport and temperature dependent chemical reactions, as described above) are missing. The impact of this approximation on our results seems to be small though, as discussed thereafter (sections 3 and 5). Note also that on timescales of the 11yr cycle, Swartz et al. (2012) found that their photolysis-only simulation captured almost all of the solar cycle effect on ozone.*"

- Discussion on results of Figure 6 (now Figure 7) has been extended as follows:

"*We now analyse the CCM ensemble results. The ensemble mean ozone sensitivity profiles (Figs. 7c and f) markedly differ with ozone sensitivity profiles derived from observations (Figs. 7a and d) and CTM (Figs. 7b and e) at the corresponding periods. These differences are particularly pronounced in the upper stratosphere (above ~5 hPa). On the other hand, despite the two different periods, the ensemble mean ozone sensitivity profiles show very similar features with positive sensitivity from 15 hPa to the stratopause and a maximum sensitivity of 0.4 at ~3 hPa (Figs. 7c and f). This maximum tropical sensitivity value and its altitude level is in good agreement with previous CCM estimates (e.g. Rozanov et al., 2006; Austin et al., 2007; Gruzdev et al., 2009; Kubin et al., 2011). The CCM ozone sensitivity analysis has also been repeated for the period 2003-2005 (not shown) to be directly comparable with the CCM results of Sukhodolov et al. (2017): we found very similar ozone sensitivity profiles.*"

- We added a full paragraph to discuss this aspect in section 5:

"*In our CCM experimental design, the direct radiative effect of UV on heating rates has been neglected leading to an underestimated temperature response to the 27day cycle. As a consequence, this may affect the ozone response by reducing the magnitude of the solar-induced temperature feedback on chemical reaction rates. Nevertheless, a detailed comparison of the ozone response in our analysis with results from previous independent CCM studies (Rozanov et al., 2006 ; Sukhodolov et al., 2017) revealed a very good correspondence, despite the fact that their*"

*experimental design included the direct radiative heating effect. This suggests that the feedback exerted by the solar-induced temperature fluctuations on ozone is modest, at least at the 27day time scales. This is in fact not surprising. In their recent study, Sukhodolov et al. (2017) shown that the atmospheric internal variability largely dominates the variability of the stratospheric temperature on 27day time scales, making the temperature response to the solar forcing difficult to identify and, hence, the influence of its feedback on ozone secondary. Nonetheless, we recognize that to quantify properly the impact of the neglected solar-induced temperature feedback on our results, additional CCM experiments including the direct radiative effect of UV on heating rate should be performed."*

- We added some details regarding the role of HOx on ozone in the upper stratosphere:

In the introduction, we added a short paragraph to describe the effect of HOx on ozone in the upper stratosphere and mesosphere: "*Fleming et al. (1995) further stressed the increasing importance with height of the solar-modulated HOx chemistry on the ozone response above 45 km. In the upper stratosphere and mesosphere, enhancement of HOx through photolysis of water vapour in Lyman-alpha line associated with an increasing solar irradiance contribute to destroy ozone. Above ~65 km and at zero-lag, the latter mechanism dominates over ozone production (i.e. by photolysis of oxygen) leading to a negative ozone-solar irradiance correlation. In the upper stratosphere and lower mesosphere (below 65 km), although ozone production dominates, increasing HOx at zero-lag contributes to the negative lag of the ozone response (Rozanov et al., 2006).*"

We also mention this mechanism in section 3.2 when we discuss the cross-correlation analysis shown in Figure 5: "*Above 3 hPa (~40 km), CCM cross-correlations of both periods (Fig. 5c,f) show a maximum at negative time lag (-2 days). As mentioned in the introductory section, this negative time lag can be induced by temperature feedback on ozone and by increasing HOx with solar irradiance which contributes to destroy ozone. While our model configuration allows to fully account for the HOx effect, the solar-induced temperature response is limited since the direct radiative heating effect is not included. The temperature response to the 27-days cycle is thus solely controlled by ozone production in the photolysis scheme. Although a temperature signal is found (not shown), it is small, reducing the likelihood for the solar-induced temperature feedback to be prominent in our experiments. Despite the approximation made in our model configuration, we notice however that the upper stratosphere negative lags compare very well with those found in CCM experiments of Sukhodolov et al. (2017) (see their Fig. 3) in which both HOx and solar-induced temperature feedback effects are fully included. Hence, this suggests that neglecting the direct effect on heating rates generated by UV variations has a limited effect on the ozone response, at least at 27-days timescale. More sensitivity experiments are required however to quantify accurately the impact of this approximation.*"

2. It is also not clear why the authors used 3 year period to evaluate ozone response from the observation and model while they show that 3 year time period does not provide statistically robust results (uncertainty only below 50%). This should be somehow explained to the readers. The choice of the number of ensemble runs is also doubtful in the light of the obtained results. If the ensemble run is crucial it would be logical to estimate how many ensemble runs are necessary to reach some kind of convergence. Moreover, the obtained results with free running CCM will be more convincing if the analysis of the CCM runs without solar rotation variability is added.

Regarding the first part of the comment (about 3 year periods), section 3 of our paper, which considers the declining phases of cycles 22 and 23, actually serves as a case study where we compare observations and model results. It follows up on Bossay et al. [2015] which found some intriguing behavior in the response of the ozone to the 27-day solar cycle. Namely, that the correlation between the ozone and the forcing seems stronger when the amplitude of solar

rotational fluctuations is small (more details are written in the introduction). In our study, we used in addition to observations, CTM and an ensemble of CCM simulations results. Section 3 allows illustrating clearly, with a concrete case, the effect of internal variability when retrieving the response of ozone to solar forcing at 27-day timescale.

To make the purpose of section 3 clearer, we revised the introductory part as follows:

*"As a first step, we follow up on the case study of Bossay et al. [2015] and make use of observations and modelling results comparison to provide a detailed picture of the ozone response to the solar rotational cycle during the declining phases of cycle 22 and cycle 23. We particularly aim to better understand the strong differences in the ozone response to solar rotational cycle found between the two periods. Two configurations of the LMDz-Reprobus chemistry climate model simulations are used, with specified dynamics (i.e. Chemistry Transport Model, or CTM) and in its free running mode (CCM). In the CTM configuration, temperature and wind fields calculated by the model are relaxed towards meteorological analysis; the dynamics is expected to be rather close to the reality, allowing direct comparisons with satellite observations for evaluating model chemical processes and its relevance to our study. In the CCM configuration, an ensemble of simulation is performed. Comparing the CCM ensemble results to CTM and observations during the declining phases of cycle 22 and cycle 23 allows to understand better the effect of internal dynamical variability on the ozone response. As a second step, we take advantage of the ensemble of CCM simulations and its large statistics to (i) assess the influence of the solar cycle phase on the ozone sensitivity to the rotational cycle and (ii) quantify the time window required for a robust estimation of the ozone sensitivity."*

We also revised the introductory of section 3. part as follows:

*"In this section, we analyse the ozone response to the solar rotational cycle over the declining phase of solar cycles 22 and 23 in the observations and in the CTM and CCM model simulations. The analysis presented here follows up on Bossay et al. (2015) observational study. In particular, we aim to assess the model performances, understand better the differences in the results between the two solar declining phase periods and highlight the importance of internal dynamical variability."*

Regarding the second point (choice of the number of ensemble members), quantifying the number of ensemble members required to reach some kind of convergent was done in Sukhodolov et al. (2017). In our study, we address a similar problem in terms of size of time window (Figure 11) where a convergence is also reached.

Finally, as suggested by the reviewer, we added a figure (now Figure 5) to compare ozone/F205 coherency estimates of the unforced vs. forced experiment results.

[Figure]

**CCM (1991–1997)**

*"Figure 5. Vertical profile of the mean squared coherence between ozone and F205 averaged between 22 and 30 day periods and calculated for the time period 1991-1997. The black lines correspond to the results of individual ensemble members (five in total) and the red lines to the results of the experiment forced with constant solar forcing. The vertical dashed line indicates the 90% confidence limit."*

We added the following part:

*"To further test the robustness of the coherence signal, we perform an additional CCM simulation for the period 1991-1997 where the solar forcing is kept constant by using fixed (i.e. climatological) photolysis rates during the model simulation. Results are shown on Fig. 5. Below 15 hPa, the different experiments show no significant coherence between ozone and solar flux. Between 15 and 1 hPa, all forced experiments (black lines) reveal a similar and significant coherence signal while for the constant solar forcing experiment (red line), the coherence is weak and within the range of randomness. The absence of significant coherence found in the constant solar experiment confirms that the coherence found between F205 and stratospheric ozone is not fortuitous and primarily originates from photolysis processes"*

Minor issues:

1. Page 7, line 1: It is clear since 2004 that two bands scheme cannot be used for the simulation of the atmospheric response to solar irradiance variability. It was confirmed in 2010,2011 and 2014. Several modifications of different complexity are known.

In its current version the LMDz-Reprobus radiative scheme has only 2 spectral bands resolved in the UV. This leads to an underestimation of the stratosphere temperature response to solar spectral variations (see also Figure 3.17 in chapter 3 of the CCMVal report). In the coming version of LMDz-Reprobus (prepared for CMIP6), the radiative scheme spectral resolution in the UV range has been improved.

2. Page 7, line 17: I do not understand how climatological temperature can be used in CCM. Please, elaborate. By the way the reference to TUV is confusing. If I am not mistaken in the cited 1999 paper the tropospheric version was described. Maybe it is better to use recent intercomparison of the codes where TUV showed excellent performance relative to reference models. I do not also understand whether or not daily NRL solar irradiance was used for the photolysis rate calculations.

In the off-line look-up tables, the temperature dependence of the absorption cross-section are computed using US standard atmosphere temperature profile in TUV, not in the CCM. This approximation has no significant impact on the results. We changed the sentence as follows: *"The temperature dependence of absorption cross-sections is calculated off-line in TUV using the US standard atmosphere."*

Regarding TUV, we now added the reference to Sukhodolov et al. (2016).

Finally, we clarified that daily NRL irradiance is used for photolysis rate calculation as follows: *"A separate photolysis look-up table is calculated every day using the daily NRLSSI as solar input"*

3. Page 7, line 21-24: I recommend to check carefully this sentence. I do not recall such a bold statement in the cited paper.

This sentence and the reference to Swartz et al. [2012] have been removed; the statements were indeed too strong. Instead, we now detail the potential effect of neglecting direct heating rates response on ozone (see also answer to comment 1):

"Thus, the solar rotational cycle forcing is taken into account by using daily photolysis rates calculated by TUV in the photochemistry module of LMDz-Reprobus. *Note however that the direct effect on heating rates generated by UV variations associated with the 27-day rotational cycle is neglected: i.e. daily changes in the spectral irradiance are not considered by the model radiative scheme. As a consequence, part of the thermal and dynamical responses to the 27-day rotational cycle and hence their effect on ozone (through transport and temperature dependent chemical reactions, as described above) are missing. The impact of this approximation on our results seems to be small though, as discussed thereafter (sections 3 and 5)."*

4. Page 12, line 9: I think this statement is not completely correct. In general, the magnitude of the rotational cycle depends on non-homogeneity of the dark/bright features distribution, which can be very small for very high level of solar activity.Fig.8 shows for example rather small variances in 2002, while the solar activity level is high.

We agree with the reviewer and modified the text as follows:

*"The amplitude of the rotational cycle depends on the inhomogeneous brightness structure of the solar disc (i.e. distribution of sunspots and faculae). Given that the amount of sunspots and faculae increases with increasing solar activity, inhomogeneity in the brightness is likely to increase during solar maximum phases. One may thus expect minimum and maximum sensitivity during 11year solar maximum and minimum phases, respectively."*

Overall, this is a valuable comparison study of the ozone response to short-term solar UV variations in both observations and a state-of-the-art chemistry climate model. The analysis is detailed and the results offer plausible explanations for differing results obtained in observations covering different time periods. Final publication is certainly expected in ACP. However, I have some important comments that will require some revision.

We thank the reviewer for reading the manuscript and providing helpful comments and suggestions. We address the raised issues in turn below.

Our answers to comments and suggestions are written in blue and manuscript changes are written in italic type within double quotes ("*like this*").

**Main Comments:**

(1) In the description of the adopted CCM configuration in section 2.3 (p. 7), the authors say: "We do not take into account the direct effect on heating rates generated by UV variations because previous modelling studies have already shown that the stratospheric ozone response to solar variations is almost entirely driven by the effects of UV changes on the photolysis rates, in particular the photolysis of molecular oxygen (Swartz et al, 2012)." Even on the 11-year time scale when a steady-state approximation is allowed and both photolysis and radiative heating are accounted for, temperature feedback reduces the ozone response in the upper stratosphere at 2 hPa by about 30% compared to that calculated by considering changes in photolysis only (see Figure 2 of Swartz et al.). 30% is still a fairly large fraction and should not be neglected. On the 27-day time scale, it is more important to include radiative effects on temperature and their feedbacks on the ozone response for two reasons. First, on this time scale, the temperature response peaks at a positive phase lag. As reviewed in the Introduction (lines 5 to 14 on p. 3), the lagged temperature response significantly alters (reduces) the ozone response and shifts it to a negative phase lag in the upper stratosphere. Second, as also reviewed there, a dynamical component of the response is produced in the upper stratosphere which feeds back into the temperature response resulting in a larger effect on the ozone response than would be predicted by a 1D radiative-photochemical model. Therefore, please modify section 2.3 to note and discuss these issues and whether the neglect of the modeled temperature response (and its accompanying dynamical response) may lead to errors in the CCM results that would not be present in simulations done in the CTM mode (forced using observed dynamics and temperatures).

We agree that neglecting the direct effect on heating rate should be considered carefully. To test the potential influence of the missing temperature feedback on ozone in our model setup, we performed additional analysis that we compared with recent model results of Sukhodolov et al. [2017] (see the answer to comment 2 for details). The manuscript has been revised at several places to discuss the effect of the neglect (see also answer to specific comments).

More specifically in section 2.3, we modified the text as follows:

"Thus, the solar rotational cycle forcing is taken into account by using daily photolysis rates calculated by TUV in the photochemistry module of LMDz-Reprobus. *Note however that the direct effect on heating rates generated by UV variations associated with the 27-day rotational cycle is neglected: i.e. daily changes in the spectral irradiance are not considered by the model radiative scheme. As a consequence, part of the thermal and dynamical responses to the 27-day rotational cycle and hence their effect on ozone (through transport and temperature dependent chemical reactions, as described above) are missing. The impact of this approximation on our results seems to be small though, as discussed thereafter (sections 3 and 5). Note also that on timescales of the*

*11yr cycle, Swartz et al. (2012) found that their photolysis-only simulation captured almost all of the solar cycle effect on ozone."*

 (2) Figure 6 compares the vertical profile of the ozone sensitivity to the solar UV (percent change in ozone for a 1 per cent change in solar UV at 205 nm) as derived from observations for two time periods, from the model using specified temperatures and dynamics (CTM), and from the model in a free-running mode (CCM). While the observational and CTM results agree fairly well, the mean CCM results show a much larger response in the upper stratosphere than is seen in either the observations or the CTM results. There is apparently no mention of this disagreement in the manuscript. In view of comment (1) above, it seems possible that part or all of the disagreement is due to neglect of the UV-induced temperature response in the CCM, which would modify both the amplitude and phase lag of the modeled ozone response. The sensitivity calculation is apparently at zero lag so it does not take into account the actual phase lag of the ozone response. Therefore, please modify the results and conclusions sections to consider the possibility that the chosen CCM configuration does not accurately simulate the net ozone response in the upper stratosphere (taking into account both the radiatively and the dynamically induced temperature response).

In a recent study, Sukhodolov et al. [2017] performed an ensemble of 30 simulations for the period 2003-2005 with the SOCOL CCM model. In their experimental setup, they considered both (i) the effects of UV changes on the photolysis rates and (ii) the direct radiative effects on temperature which feeds back on ozone. Hence, their CCM formulation allows accounting for all photochemical and radiative effects. Their large number of simulations and the fact that they simulated the period 2003-2005 using NRLSSI data (like us) allows comparing with our CCM ensemble results in a fair way. Their UV-ozone cross-correlation analysis (see their Figure 3b) reveal a negative lag of 1 to 2 days in the upper stratosphere (above 3 hPa) which is consistent with the one we found in our study (see our Fig. 7c,f). When performing the UV-ozone cross-correlation analysis over the same period (i.e. 2003-2005), we also found a negative lag of 1 to 2 days. We also compared the analysis of the ozone sensitivity to F205 index (see below) over the period 2003-2005. Their ozone sensitivity profiles (left) are shown at optimum lag and ours (right) at optimum lag (solid) and zero lag (dashed). Results from both models compare very well, showing a maximum slightly larger than 0.4 at 3-4 hPa and decreases down to 0.25 near the stratopause. The very good agreement between the two CCMs, despite their different formulation, suggest that omitting the direct radiative heating does not have a strong effect on the ozone response and is therefore an acceptable approximation in our case. Consequently, this model comparison suggests that the disagreement between the model and the observations in the upper stratosphere most likely comes from the dynamical variability, in line with our former interpretation. In 1991-1994, observational uncertainties (either in MLS or in the reanalysis used to nudge the model) may have an influence on the results. We will discuss this thoroughly in the revised version of the manuscript.

Nonetheless, we recognize that, ideally, the direct radiative heating should be considered. In its current version the LMDz-Reprobus radiative scheme has only 2 spectral bands resolved in the UV. Hence, this would lead anyway to an underestimation of the stratosphere temperature response to solar spectral variations (see also Figure 3.17 in chapter 3 of the CCMVal report). For the next version of LMDz-Reprobus, the radiative scheme spectral resolution in the UV range has been improved.

[Figure]

Globally, results and conclusions of the manuscript have been revised to discuss the CCM configuration => see answer to specific comments below (particularly comments 13, 15, 17, 18).

Specifically, discussion on results of Figure 6 (now Figure 7) has been extended as follows:

*"We now analyse the CCM ensemble results. The ensemble mean ozone sensitivity profiles (Figs. 7c and f) markedly differ with ozone sensitivity profiles derived from observations (Figs. 7a and d) and CTM (Figs. 7b and e) at the corresponding periods. These differences are particularly pronounced in the upper stratosphere (above ~5 hPa). On the other hand, despite the two different periods, the ensemble mean ozone sensitivity profiles show very similar features with positive sensitivity from 15 hPa to the stratopause and a maximum sensitivity of 0.4 at ~3 hPa (Figs. 7c and f). This maximum tropical sensitivity value and its altitude level is in good agreement with previous CCM estimates (e.g. Rozanov et al., 2006; Austin et al., 2007; Gruzdev et al., 2009; Kubin et al., 2011). The CCM ozone sensitivity analysis has also been repeated for the period 2003-2005 (not shown) to be directly comparable with the CCM results of Sukhodolov et al. (2017): we found very similar ozone sensitivity profiles."*

**Other Comments:**

(3) Introduction, first paragraph, last sentence. "A thorough understanding and accurate quantification of the UV variability effect on the middle stratosphere from which the "top-down" theory stems, are thus necessary." If so, then why is the CCM configuration limited to only the photochemical ozone response? The thermal response and its associated dynamical response are the main components of the top-down mechanism for solar influences on the troposphere.

Indeed, our study examines the photochemical ozone response, not the more general thermal and dynamical responses. We modified the manuscript as follows:

*"A thorough understanding and accurate quantification of the UV variability effect on the middle stratosphere ozone are thus necessary."*

(4) Section 2.1, line 11. Are you using the NRL SSI version 1 or version 2? It is fine if you are still using version 1 but it should be clarified. Version 2 is available from https://data.noaa.gov/dataset/noaa-climate-data-record-cdr-of-solar-spectral-irradiance-ssi-nrlssi-version-2

We are indeed using the NRL SSI model version 1. This is now clarified in the revised version of the manuscript in section 2.1:

*"In our study, we use the solar spectral irradiance provided by the Naval Research Laboratory Solar Spectral Irradiance (NRLSSI) model version 1 (Lean, 2000; Wang et al., 2005)."*

(5) Section 2.2, line 24. Please specify the pressure levels for ozone retrievals for the two MLS instruments. Which versions of the UARS MLS and AURA MLS data sets are being used for this analysis? Please reference more up-to-date descriptions of these data. The Version 5 UARS MLS data set is described by Livesey et al., JGR, v. 108, doi:10.1029/2002JD002273, 2003. Please give URLs where readers who wish to repeat the analysis can download the data. For example, the UARS MLS data are at https://mls.jpl.nasa.gov/uars/data.php.

We now clarified the pressure levels for ozone retrievals for versions 5 of UARS MLS and 4.2 of AURA MLS that are used in our study and give the references of the more up-to-date descriptions of the corresponding data (Livesey et al. 2003 for UARS MLS and Livesey et al. 2017 for AURA MLS). The URLs where data can be accessed are provided at the end of the paper section 6 (data availability).

Changes in the manuscript have been made in section 2.2:

*"We used the version 5 UARS MLS dataset described Livesey et al., (2003). The ozone retrieval is based on 205 GHz radiances, provided onto 13 pressure levels in the range 100-1 hPa (100, 68.1, 46.4, 31.6, 21.5, 14.7, 10, 6.8, 4.6, 3.2, 2.2, 1.5 and 1 hPa) and has an average vertical resolution of 4 km in the stratosphere. The typical 1σ precision for ozone mixing ratio measurements is ~0.3 ppmv between 68 and 1 hPa."*

(6) Section 2.2, line 31. If only 30% of the measurements are in the daytime, another problem arises, which is the ozone diurnal cycle. This cycle becomes important at roughly 2 hPa and above. Including 70% of measurements at night will therefore have the effect of reducing the estimated ozone response to solar UV variations at 2 hPa and above. This will not affect comparisons with the CTM and CCM provided that the model "measurements" also include both day and night data. Ideally, there should be 70% night and 30% day model data to allow an exact comparison. Please add text to explain this.

[Figure]

We tested the influence of an uneven distribution of night/day time measurements (see figure above). To do so, we reproduced the 2004-07 analysis Aura-MLS data but by resampling the data

with the ratio of 70/30 of night/day time measurements and compared it to the daily average (which has a ratio of roughly 50/50). The results are fairly similar, indicating a limited diurnal effect on the unbalanced day/night sampling. Note that we repeated the analysis by considering only nighttime or daytime measurements (not shown) and did not find any significant difference.

The ozone diurnal cycle issue is now mentioned in section 2.2 :

*"Furthermore, the ozone diurnal cycle becomes important in the upper stratosphere, so that the results may be affected by the imbalance in daytime and night-time measurements used to construct daily time series. This issue will be discussed in section 3.2."*

(7) Section 2.3, line 24. 39 levels and 70 km lid means a resolution of less than 2 km. This is much better than the MLS vertical resolution, which is about 6 km. One should mention this before making direct comparisons in the following sections.

As clarified previously, the UARS MLS (Aura MLS) vertical resolution for ozone retrieval is 4 (3) km in the stratosphere. In the LMDz-Reprobus, model, the vertical resolution slowly varies with altitude, from 1 km in the upper troposphere to 3 km in the middle and upper stratosphere (i.e. above 10 hPa). Between 10 hPa and 1 hPa – i.e. the region we are focusing on - MLS and LMDz-Reprobus have roughly the same vertical resolution (~3 km).

We now clarify this in section 2.3:

"The model uses a classical hybrid σ-P coordinate in the vertical, has 39 vertical levels and a lid-height at ~70 km. *The model vertical resolution slowly decreases with height. In the middle and upper stratosphere (30-50 km or ~10-1 hPa) - focus of our study – the model vertical resolution reaches 3 km which is similar to the vertical resolution of UARS-MLS and Aura-MLS measurements in this altitude range. The model is integrated with a horizontal resolution of 3.75° in longitude and 1.9° in latitude.* The equations are discretized on a staggered and stretched latitude-longitude Arakawa-C grid."

(8) Figure 1. The units should be $W/m_2/nm$.

This has been corrected.

(9) Section 3.2, Figure 3. The periodogram of the MLS ozone measurements (Figure 3a,d) is done at 3.2 hPa. But, according to Livesey et al. (2003), the UARS MLS measurements were not retrieved at this level, only at 2.2 and 4.6 hPa. So, how are data obtained at 3.2 hPa?

For the version 5 of the UARS MLS measurements which is used in our study, the vertical retrieval grid over the stratosphere and lower mesosphere has been doubled compared to previous versions (i.e. v4 and before). Hence, the ozone measurements are provided at levels 2.2, 3.2, and 4.6 and not only 2.2 and 4.6 (see also Table 6 in Livesey et al. 2003). See also answer to comment #5.

(10) Section 3.2, lines 10-12. Please note that the lack of an obvious solar rotational signal in the MLS data considered here is partly because the measurements were obtained during the declining phases of solar activity using a limb sounding instrument, whose measurements are spatially and temporally sparse. The ozone signal is more easily detectable and repeatable in daily zonal means of nadir-viewing backscattered ultraviolet measurements under solar maximum conditions when solar UV variations are stronger and more coherent. The CTM simulations are also affected by the relatively weak solar rotational UV variations during the selected time periods.

We agree that the solar rotational signal should be more easily detectable during maximum phases of solar activity than during declining phases and this may contribute to the difficulty in the identification of a prominent peak in the power spectrum. Note however that we repeated the MLS power spectrum analysis during the maximum phase of solar cycle 24 (2012-2015) and we still did

not identify a clear peak (over this period, the MLS sampling is large since almost 800 profiles are retrieved each day). Regarding daily zonal means of nadir-viewing backscattered ultraviolet measurements (like SBUV), it may be indeed easier to detect but we did not find any reference where this is clearly shown in power spectrum analysis (only in coherency as in our study).

We modified section 3.2 accordingly:

*"This illustrates the difficulty in detecting solar rotational signals in the observations, as well as in a single ensemble member over these 3year periods. Note that we additionally computed periodograms in observations during solar maximum phases (i.e. 2012-2015) where 27day fluctuations in the solar forcing are stronger than during the declining phase (not shown). The results were however similar and no clear peak at 27 days could be identified. Hence, the absence of a distinctive rotational signal suggests the presence of strong and rather random ozone variability of non-solar origin which makes the ozone rotational signal very difficult to detect and estimate."*

(11) P. 9, Figure 4. Normally, a cross-spectral analysis should yield phase estimates as well as coherency estimates. There is no mention of phase on p. 9 so it must be assumed that the coherency estimates are at zero lag. But the cross-correlation functions in Figure 5 show that the phase lags are not constant with altitude and are not always zero. They tend to be somewhat negative in the upper stratosphere and become positive in the middle and lower stratosphere. The ozone-UV sensitivities shown in Figure 6 are also presumably at zero lag. This differs from previous observational studies (e.g., Hood and Zhou, 1998) which calculated sensitivities at the so-called optimum lag, i.e., the lag where the correlation maximizes. Please add text to explain that these calculations are being done at zero lag and why this lag is chosen.

We made the choice not to provide the phase lag with the coherency since we calculate the cross-correlation afterwards which provides basically the same information and both are consistent.

The sensitivities are indeed shown at lag 0 and not at optimum lags. Optimum lags are in fact not simple to define as they may vary between observations and models results, between two different periods of observations, or between two ensemble members. Alternatively, we could choose one reference optimum lag vertical profile upon which the sensitivity would be calculated. But similarly, this poses the problem of defining the most accurate reference profile; shall we use observational or model results? Hence we opt for the lag 0 as a common reference. Finally, note that we tried both but it did not affect the results and conclusions as the sensitivity profiles shown in the answer to comment #2 reveal.

Section 3.2 has been modified accordingly:

*"In previous studies, ozone sensitivity profiles were either calculated at optimum lags where the correlation coefficient maximizes (e.g. Hood and Zhou, 1998) or at zero lag (e.g. William et al., 2001; Austin et al., 2007). Both alternatives were tried but given the limited effect on the results and conclusions, we elected to show only ozone sensitivity profiles using a common time frame, hence at zero lag. Results are shown on Fig. 7."*

(12) P. 10, line 24. Typo: Seizing? Caption to Figure 3: from the runs ensemble?

We changed "*Seizing*" for "*Marked*" and corrected caption 3: "*The middle panels (b and e) represent the ozone Lomb-Scargle periodograms for CTM simulation and the bottom panels (c and f) the average periodogram of the CCM ensemble. The dotted envelop (c and f) indicates the 2σ standard deviation of the ensemble of CCM simulations.*"

(13) P. 11, top of page. The CCM results shown in Figure 5c,f are characterized by negative lags near the stratopause. What is the cause of these negative lags? Is it feedback from a temperature response caused only by increased radiative heating associated with the ozone response (holding

direct UV heating changes constant)? Or, is it increased photolysis of water vapor in the lower mesosphere and resulting destruction of ozone by odd hydrogen? Or both? Can this be diagnosed?

Below are shown the cross-correlation analysis (as in Fig. 5) for the hydroxyl and the temperature for the CCM ensemble mean. The hydroxyl response is highly consistent with the results of Sukhodolov et al. (2017) (see their Figure 3). The temperature response is also consistent. Thus even in absence of the direct UV heating, an "indirect" temperature response is produced by the ozone response. These two effects may thus contribute to the negative lags near the stratopause. Note the particularly good agreement in the upper stratosphere ozone phase lag with Sukhodolov et al. (2017) which suggests that the OH contribution is more important than the temperature one because in our case the temperature response is strongly reduced: the temperature sensitivity in our experiments yields a maximum of 0.035 in the upper stratosphere instead of 0.1 in Sukhodolov et al. (2017).

[Figure]

We modified the manuscript as follows:

- In the introduction, we added a short paragraph to describe the effect of HOx on ozone in the upper stratosphere and mesosphere: "*Fleming et al. (1995) further stressed the increasing importance with height of the solar-modulated HOx chemistry on the ozone response above 45 km. In the upper stratosphere and mesosphere, enhancement of HOx through photolysis of water vapour in Lyman-alpha line associated with an increasing solar irradiance contribute to destroy ozone. Above ~65 km and at zero-lag, the latter mechanism dominates over ozone production (i.e. by photolysis of oxygen) leading to a negative ozone-solar irradiance correlation. In the upper stratosphere and lower mesosphere (below 65 km), although ozone production dominates, increasing HOx at zero-lag contributes to the negative lag of the ozone response (Rozanov et al., 2006).*"
- We also mention this mechanism in section 3.2 when we discuss the cross-correlation analysis shown in Figure 5: "*Above 3 hPa (~40 km), CCM cross-correlations of both periods (Fig. 5c,f) show a maximum at negative time lag (-2 days). As mentioned in the introductory section, this negative time lag can be induced by temperature feedback on ozone and by increasing HOx with solar irradiance which contributes to destroy ozone. While our model configuration allows to fully account for the HOx effect, the solar-induced temperature response is limited since the direct radiative heating effect is not included. The temperature response to the 27-days cycle is thus solely controlled by ozone production in the photolysis scheme. Although a temperature signal is found (not shown), it is small, reducing the*"

*likelihood for the solar-induced temperature feedback to be prominent in our experiments. Despite the approximation made in our model configuration, we notice however that the upper stratosphere negative lags compare very well with those found in CCM experiments of Sukhodolov et al. (2017) (see their Fig. 3) in which both HOx and solar-induced temperature feedback effects are fully included. Hence, this suggests that neglecting the direct effect on heating rates generated by UV variations has a limited effect on the ozone response, at least at 27-days timescale. More sensitivity experiments are required however to quantify accurately the impact of this approximation.”*

(14) P. 11, bottom of page. In addition to not mentioning the anomalously large CCM response in the upper stratosphere, there is also no mention here of the likely effect of the ozone diurnal cycle in reducing the ozone response in the upper stratosphere relative to that measured earlier from backscattered ultraviolet instruments, which operated only in the daytime. This difference is emphasized in Hood and Zhou [1998] for example.

The CCM response is indeed anomalously large compared to observations and CTM results. It is however not anomalously large when comparing with Sukhodolov et al. (2017) (see also answer to comment 2). Section 3.2 has been modified accordingly.

Regarding the diurnal cycle, we tested its influence on the results over the period covered by Aura-MLS (i.e. 2004-2007). The Aura-MLS instrument measures in the tropics at one fixed night local time (~0142 LST) and one fixed day local time (~1342 LST). To mimic an irregular sampling with respect to the local time, we repeated the Aura-MLS analysis as follows:

1. We initially build the ozone time series using daytime measurements only (1095 days in total)
2. Out of these 1095 days, we select N days randomly where daytime measurements are replaced by nighttime measurements only.
3. We then compute the ozone sensitivity.

The procedure was repeated for N=100 (i.e. 91% of daytime measurements), N=500 and N=1000 (i.e. 9% of daytime measurements) (see figure below) and the results reveal very minor differences between the various sensitivity profiles. Hence this analysis suggests that the diurnal cycle has a small effect on the ozone solar rotational signal.

[Figure]

We added a paragraph to discuss ozone diurnal cycle at the end of section 3.2:

*“As mentioned in the section 2.2, the results based on UARS-MLS measurements may be affected by the imbalance between night and daytime sampling due to the ozone diurnal cycle becoming significant in the upper stratosphere. To test the influence of the ozone diurnal cycle, we repeated all the analysis performed in this section by mimicking an irregular sampling over the period covered by Aura-MLS (i.e. 2004-2007). Each day, ~800 ozone vertical profiles of the Aura-MLS instrument are evenly retrieved in the tropics [20S-20N] at two fixed local times: one at night (~0142 LST) and*

*one during daytime (~1342 LST). We initially build the ozone time series using daytime measurements only (1095 days in total). Among these 1095 days, we selected N days randomly where daytime measurements were replaced by night time measurements. We then repeated the spectral, correlation and regression analysis. The procedure was performed for various values of N, from N=100 (i.e. 91% of daytime measurements) to N=1000 (i.e. 9% of daytime measurements). The results (not shown) revealed almost no dependence to N, suggesting that the diurnal cycle has a small effect on the ozone solar rotational signal.*"

(15) Section 4. While it is useful to carry out these analyses, one must question whether the CCM in its chosen configuration (no direct solar UV heating changes) is ideal for this purpose. Also, a time window with a length of 10 years includes both solar maximum periods (when 27-day UV variations are strong and numerous) as well as solar minimum periods (when these variations are weak and sparse). Could it therefore be possible that a shorter time window of 3 years centered on a strong solar maximum (e.g., that in 1979-82) could yield more reliable results than a 10-year window which includes mostly non-maximum solar conditions?

Regarding the suitability of the chosen configuration (see also answer to major comment #2), paragraphs have now been added at several places in section 3.2 where we discuss our results by comparing with previous independent CCM studies (i.e. Rozanov et al., 2006 ; Sukhodolov et al., 2017). We further mention this in the concluding paragraph of section 3:

"*Moreover, the fine correspondence of our results with those based on independent previous chemistry-climate modelling experiments (e.g. Rozanov et al., 2006; Sukhodolov et al., 2017) emphasizes the relevance of our experimental model setup (i.e. despite neglecting the direct effect on heating rates) to examine the ozone response to 27day solar variations.*"

Regarding the discussion of the length of the time window, we agree that it strongly depends on the amplitude of solar rotational fluctuations and, hence, the phase of the solar cycle. We examined this further with the solar maximum of cycle 23 with the model and found that choosing a 10year time window length still leads to the more accurate and robust estimates of the ozone sensitivity in comparison with restricting the analysis to one solar maximum period only. Of course, the strongest the solar maximum and its associated 27day solar fluctuations are, the shorter the time window size can be. But if a best option has to be given, our results seem to suggest that a long time window is preferable.

We modified the main text of section 4 to make this point clearer:

"*These uncertainty ranges also strongly depend on the amplitude of the solar rotational variations and hence the phase of the 11year solar cycle; we find that during solar maximum of cycle 23, minimum of cycle 22, a minimum time window size of 2, 5 years, respectively, is required for the standard deviation to drop under 50%. To obtain a standard deviation lower than 20%, we however found that randomly choosing a 10year time window length performs better than restricting the analysis to short but solar maximum period only (i.e. solar 23). These results suggest that long time series are preferable to estimate accurately the ozone sensitivity to solar rotational fluctuations in observations.*"

(16) Minor corrections: In the abstract, lines 23-24, neither nor should be either or. P.13, line 10. anti-correlation should be inverse correlation.

This has been corrected.

(17) P. 15, lines 11,12: "Applying the same spectral analysis to the average of the CCM ensemble simulations allows reducing the 'masking' effect by random dynamical variability, so that the rotational signal in ozone can be more easily identified and estimated." However, the negative aspect of this approach is that the CCM may not perfectly simulate the actual ozone response to

short-term UV variations, partly because of the neglect of the direct radiative effet of the UV variations in the model, and their secondary dynamical effects.

We added a full paragraph to discuss this aspect in section 5:

"*In our CCM experimental design, the direct radiative effect of UV on heating rates has been neglected leading to an underestimated temperature response to the 27day cycle. As a consequence, this may affect the ozone response by reducing the magnitude of the solar-induced temperature feedback on chemical reaction rates. Nevertheless, a detailed comparison of the ozone response in our analysis with results from previous independent CCM studies (Rozanov et al., 2006 ; Sukhodolov et al., 2017) revealed a very good correspondence, despite the fact that their experimental design included the direct radiative heating effect. This suggests that the feedback exerted by the solar-induced temperature fluctuations on ozone is modest, at least at the 27day time scales. This is in fact not surprising. In their recent study, Sukhodolov et al. (2017) shown that the atmospheric internal variability largely dominates the variability of the stratospheric temperature on 27day time scales, making the temperature response to the solar forcing difficult to identify and, hence, the influence of its feedback on ozone secondary. Nonetheless, we recognize that to quantify properly the impact of the neglected solar-induced temperature feedback on our results, additional CCM experiments including the direct radiative effect of UV on heating rate should be performed.*"

(18) P. 15, lines 18-21: "Analysis of the CCM ensemble simulations suggest that the differences mostly originate from the dynamical variability." Usually, internal dynamical variability in a model is larger than in observations so it is not clear that a single model run is equivalent to a single sample of observations (or a single run of the CTM). The large spread in the ensemble mean sensitivity profile could also reflect a less complete simulation of the upper stratospheric dynamical response to short-term solar UV variations.

See answer to comment #17.

Anonymous Referee #3

The paper uses models and satellite observations to investigate the response of tropical stratospheric ozone to short-term solar UV variations due to the 27-day solar rotation. This is a topic that has been investigated in a number of previous studies but is still not resolved; therefore an updated study will be of interest to journal readers. The stated goals of the present paper are to "(i) assess the influence of the solar cycle phase on the ozone sensitivity to the rotational cycle and (ii) quantify the time window required for a robust estimation of the ozone sensitivity".

While the paper does address the stated topics, the focus is drawn elsewhere by awkward organization and extraneous material. The authors do not make it clear why they include Section 3, which is longer than the section addressing their goals. Section 3 does not contribute to the focus of the paper and, in my view, does not contribute to the understanding of the solar response.

My most serious concern about the paper is in regard to the implicit assumptions in the text. At a number of places, the authors have assumed that a solar response is present even when they do not see a signal of it in their analysis or that the response is larger than what they find from the analysis. The text then describes why and how the signal has been masked. This is dangerous; if you do not find a signal of a response, the first explanation should be that there is no response. Even if processes that might mask a signal are present, it is not appropriate to conclude that this masking is the reason for not finding a signal that is "known" to be present based on prior assumptions. A more appropriate way to say it would be: if there is a response, it is too weak to detect.

We thank the reviewer for reading the manuscript and providing helpful comments and suggestions. We address the raised issues in turn below.

Our answers to comments and suggestions are written in blue and manuscript changes are written in italic type within double quotes ("*like this*").

Section 3 is a case study which follows up on Bossay et al. [2015] study using, in addition to observations, CTM and an ensemble of CCM simulations results. This part allows illustrating clearly, with a concrete case, the effect of internal variability when retrieving the response of ozone to solar forcing at 27-day timescale. Another interesting point of Section 3 is that it allows comparing model and observations and validating the model in light of the very good agreement when comparing the CTM and observational results in 2004-2007. We however agree that in the initial version of the manuscript, Section 3 was not sufficiently motivated. The manuscript has now been revised accordingly (See answer to major comment 1 below).

Regarding the second main concern of the reviewer, we agree that more caution is required when describing results where no signal is found (namely Figure 3 where a power spectrum analysis of raw ozone time series is performed). The manuscript has been revised accordingly.

The description of figure 3 and 4 has been changed in order not to presume of a solar rotational signal:

"The two periodograms of MLS ozone measurements (Fig. 3a and Fig. 3d) reveal no prominent peak in the range of the 20-30 days period, *suggesting an absence of a solar rotational signal in ozone*. More prominent peaks are found at longer periods although they are not consistent between the two periods. The large peak found at the 35day period for 1991-94 corresponds to the yaw-maneuver period of the MLS instrument as described previously (Froidevaux et al., 1994; Hood and Zhou, 1998). Similarly to observations, the periodograms of CTM results (Fig. 3b and Fig. 3e) do also not exhibit a distinctive solar rotational peak; there are some minor peaks between 20 and 30 days and their amplitudes are smaller in 2004-07 than in 1991-94. *The analysis has been repeated at lower pressure-height levels (e.g. 10 hPa, not shown) and led to the same conclusions. Overall, the raw power spectrum analysis of observations and CTM results in the middle and upper tropical*

*stratosphere does not allow identifying an ozone signal associated with the solar forcing fluctuations at rotational timescales for the two periods considered here.*

[…]

We further examine the relationship between stratospheric ozone and solar rotational cycle by performing cross-spectrum analysis between stratospheric ozone and F205. *Despite the absence of a solar rotational peak in the ozone power spectrum derived from observations and CTM results*, cross-spectrum analysis should help identifying coherent variability modes between the solar forcing and tropical ozone. Figure 4 presents the vertical profile of the magnitude-squared coherence (hereinafter referred as coherence) between F205 and tropical stratospheric ozone from MLS observations (a and d), CTM model results (b and e) and CCM model results (c and f)."

We also made correction throughout the manuscript to avoid overstatements regarding the ozone response to F205 when this is not so clear.

It is however very clear that there is a response, as the cross-spectrum analysis reveals. In this regard, we now added a new Figure to the manuscript to show the difference in the ozone/F205 coherency between forced and unforced CCM experiments.

**Major Comments**

1. As indicated above, I did not see the purpose of Section 3. Since you consider two fairly short 3-year periods, the analyses do not have any bearing on the questions raised about variations of the response with timing within the 11-year solar cycle or the dependence on the length of the analysis period.

Given that our analysis focuses on the 27-day cycle, one would expect 3 years to be sufficient to characterize the ozone response on rotational timescales because it represents about 40 cycles. Our results show it is not sufficient.

The first part of our paper, which considers the declining phases of cycles 22 and 23, actually serves as a case study where we compare observations and model results. It is a follow-up of the study of *Bossay et al*. [2015] which found some intriguing behavior in the response of the ozone to the 27-day solar cycle. Namely, that the correlation between the ozone and the forcing seems stronger when the amplitude of solar rotational fluctuations is small (more details are written in the introduction). We also found in the literature large differences between observational studies. To understand better this apparent contradiction and the variability of the ozone response to solar rotational fluctuations found in different observational studies, we need more than only one realization in order to improve the statistics and better understand the role of other sources of ozone variability. We thus used an ensemble of CCM simulations.

To make the purpose of section 3 clearer, we revised the introductory part as follows:

*"As a first step, we follow up on the case study of Bossay et al. [2015] and make use of observations and modelling results comparison to provide a detailed picture of the ozone response to the solar rotational cycle during the declining phases of cycle 22 and cycle 23. We particularly aim to better understand the strong differences in the ozone response to solar rotational cycle found between the two periods. Two configurations of the LMDz-Reprobus chemistry climate model simulations are used, with specified dynamics (i.e. Chemistry Transport Model, or CTM) and in its free running mode (CCM). In the CTM configuration, temperature and wind fields calculated by the model are relaxed towards meteorological analysis; the dynamics is expected to be rather close to the reality, allowing direct comparisons with satellite observations for evaluating model chemical processes and its relevance to our study. In the CCM configuration, an ensemble of simulation is performed. Comparing the CCM ensemble results to CTM and observations during the declining phases of cycle*

*22 and cycle 23 allows to understand better the effect of internal dynamical variability on the ozone response. As a second step, we take advantage of the ensemble of CCM simulations and its large statistics to (i) assess the influence of the solar cycle phase on the ozone sensitivity to the rotational cycle and (ii) quantify the time window required for a robust estimation of the ozone sensitivity."*

We also revised the introductory of section 3. part as follows:

*"In this section, we analyse the ozone response to the solar rotational cycle over the declining phase of solar cycles 22 and 23 in the observations and in the CTM and CCM model simulations. The analysis presented here follows up on Bossay et al. (2015) observational study. In particular, we aim to assess the model performances, understand better the differences in the results between the two solar declining phase periods and highlight the importance of internal dynamical variability."*

2. Perhaps the response diagnosed from MLS is intended as validation for your model simulations. In this case, why ignore the 9.5 years of MLS/Aura observations that have been taken since 08/2007? As you later find using model simulations, a 3-year period does not give results that are very robust and so is not convincing as validation.

In light of the observation/CTM results comparison in 2004-2007, we argue that it is rather convincing as validation: model and observations show almost the same results on the several diagnostics we applied. We also performed extra analysis over the 2003-2005 period to compare with Sukhodolov et al. [2017] which also reveal very consistent results. This is now clarified in the Section 3 of the manuscript.

In our study, it is very important to use short periods as it is what was done in previous efforts. We can thus compare our results more easily and provide explanation on the different discrepancies that have been found in these previous studies.

Further note that the prolonged solar minimum after 2007 which lasted for almost 5 years has an exceptionally weak 27-day component. As a consequence, it is not the most appropriate period to examine ozone response to 27-day solar flux variations. There is again another maximum from 2011-onwards that could be investigated and we did it but it does not bring any further elements. In this regard, we do not intend to extend our CTM and CCM ensemble simulations by ~10 years that would turn out being very expensive.

3. It seems that both satellite data (in Section 3) and model output (throughout) are zonally and latitudinally averaged over each day, including both day and night. There is a mention of local time issues (Section 2.2) but you then decide not to use any local time or day/night information in your analysis. There is evidence that this should be considered: 1) why else would the MLS/UARS ozone variations show a prominent peak at the yaw period?; 2) it is known that local time variations in the response of ozone in the upper stratosphere to solar variability are not negligible (e.g. Li et al., Earth and Space Science, doi:10.1002/2016EA000199).

[Figure]

Daily average        70%night/30%day

We tested the influence of an uneven distribution of night/day time measurements (see figure above). To do so, we reproduced the 2004-07 analysis Aura-MLS data but by resampling the data with the ratio of 70/30 of night/day time measurements and compared it to the daily average (which has a ratio of roughly 50/50). The results are fairly similar, indicating a limited diurnal effect on the unbalanced day/night sampling. Note that we repeated the analysis by considering only nighttime or daytime measurements (not shown) and did not find any significant difference.

The ozone diurnal cycle issue is now mentioned in section 2.2:

*"Furthermore, the ozone diurnal cycle becomes important in the upper stratosphere, so that the results may be affected by the imbalance in daytime and night-time measurements used to construct daily time series. This issue will be discussed in section 3.2."*

The following paragraph has been added in section 3.2:

*"As mentioned in the section 2.2, the results based on UARS-MLS measurements may be affected by the imbalance between night and daytime sampling due to the ozone diurnal cycle becoming significant above 2 hPa. To test the influence of the ozone diurnal cycle, we repeated all the analysis performed in this section by mimicking an irregular sampling over the period covered by Aura-MLS (i.e. 2004-2007). Each day, ~800 ozone vertical profiles of the Aura-MLS instrument are evenly retrieved in the tropics [20S-20N] at two fixed local times: one at night (~0142 LST) and one during daytime (~1342 LST). We initially build the ozone timeseries using daytime measurements only (1095 days in total). Among these 1095 days, we selected N days randomly where daytime measurements were replaced by nighttime measurements. We then repeated the spectral, correlation and regression analysis. The procedure was performed for various values of N, from N=100 (i.e. 91% of daytime measurements) to N=1000 (i.e. 9% of daytime measurements). The results (not shown) revealed almost no dependence to N, suggesting that the diurnal cycle has a small effect on the ozone solar rotational signal."*

Regarding the Li et al (2016) paper, the problematic is quite different. The orbit of SBUV drifts relatively to the diurnal cycle on the same timescale as the solar signal which is investigated

(decadal). So, in this case, there is an artificial decadal fluctuation simply created by the drift which aliases the "real" decadal signal of solar origin. This type of problem does not apply in our case, though.

In UARS-MLS, the yaw maneuver creates an artificial periodicity of 36 days in zonally average data. Furthermore, the non-fixed local time measurements may introduce spurious variations in the temporal evolution of the daily zonal mean (due to the diurnal cycle) in the upper stratosphere. This may partly affect the observed signals in the upper stratosphere for the period 1991-94.

4. Although some spread should be expected, if I see a response peaking at 22 days (as you show in Figure 4a), I would automatically assume that it has no relation to the solar rotation. The signal in that particular panel is near zero at 27 days. It seems shaky to interpret it as driven by the solar flux variations. Can you provide more justification for your interpretation?

Active regions are not always located at the same longitude on the Sun. Furthermore, the Sun's rotational period depends on the latitude (i.e. differential rotation). For these reasons, we do not expect a thin peak at 27 days, but a rather broad peak near 27 days. Wavelet analysis of solar forcing time series (Figure 3c) shows that solar forcing fluctuations are strong in the band 20-30 days, hence it is not shocking to see a large patch of coherency between 20 and 27 days. The most convincing evidence of a link between the solar flux variations and the ozone response is the high coherency between both: it means that the UV forcing and ozone vary together.

See also answer to the next comment (#5) for further justification.

5. ("additional CCM simulation where the solar flux is kept constant") Since you show that the CCM responses vary considerably between realizations, a single simulation is not a very useful. Alternatives would be to perform additional realizations and/or to perform a similar case (fixed solar flux) with the CTM, particularly for a longer period (>10 years).

The CCM realization with constant solar flux is performed in the CCM configuration from 1/1/1991 to 30/11/1997. To be fair, we compared the coherency in this experiment with the coherency in the five individual ensemble members over the same period. Below, we plotted the vertical profiles of the spectral coherency around 27 days between the F205 and ozone for the (dashed) constant solar experiment and (solid) the five individual ensemble members. It is clear that although the spectral coherency varies among the different ensemble members, the results are extremely similar above the 10 hPa level and very robust. If we remove the forcing, there is no coherency signal (dashed line). Given the very consistent results between the five different ensemble members, we are confident in saying that the coherency signal is due to the forcing and is not fortuitous. We now modified the manuscript as follows.

We added the following Figure:

[Figure]

*"Figure 5. Vertical profile of the mean squared coherence between ozone and F205 averaged between 22 and 30 day periods and calculated for the time period 1991-1997. The black lines correspond to the results of individual ensemble members (five in total) and the red lines to the results of the experiment forced with constant solar forcing. The vertical dashed line indicates the 90% confidence limit."*

We added the following part in the main text:

*"To further test the robustness of the coherence signal, we perform an additional CCM simulation for the period 1991-1997 where the solar forcing is kept constant by using fixed (i.e. climatological) photolysis rates during the model simulation. Results are shown on Fig. 5. Below 15 hPa, the different experiments show no significant coherence between ozone and solar flux. Between 15 and 1 hPa, all forced experiments (black lines) reveal a similar and significant coherence signal while for the constant solar forcing experiment (red line), the coherence is weak and within the range of randomness. The absence of significant coherence found in the constant solar experiment confirms that the coherence found between F205 and stratospheric ozone is not fortuitous and primarily originates from photolysis processes"*

6. I am having trouble reconciling Figure 4c, f with Figure 6 and 7. Figure 4 indicates largest signal at ~10 hPa and near zero signal at ~0.3 hPa while Figures 6-7 indicates the opposite. Even though sensitivity (Fig 6-7) is different from absolute response (Fig4), these do not appear to be consistent. Please explain.

Please note that:

1/ none of our analysis goes up to the lower mesosphere (i.e. 0.3 hPa). We will however assume that the reviewer meant 3 hPa.

2/ in the upper stratosphere, there is not near zero coherency signal at 3 hPa in Figure 4c and f (the signal is just weaker and only in Figure 4f).

[Figure]

CCM (2003–2005)

On Figure above, we plotted the ensemble mean and spread of the ozone sensitivity to F205 index over the period 2003-2005 to compare our results with those of Sukhodolov et al. (2017) (see also answer to reviewer #2). The ozone is shown at optimum lag (where the correlation maximizes, solid) and zero lag (as in our analysis throughout the paper, dashed). This analysis shows that - particularly clearly when we examine the optimum lag sensitivity profile – the sensitivity does not maximize at 10 hPa, but it corresponds to the pressure level where the sensitivity average value is the most stable (as revealed by the strongly reduced standard deviation). Hence, coherency results and sensitivity are consistent.

7. Your conclusion (p. 15, l. 18) that "the differences mostly originate from the dynamical variability" is an important one that should be brought out more prominently.

We insist on this conclusion at many places in the manuscript (abstract, results part, conclusion). The various corrections in the manuscript also allow insisting further on the importance of the internal variability.

**Specific Comments**

1. (p. 2) "the life span of a single satellite instrument is generally far less than one solar cycle" This has been true for a few but just as many last for a time span comparable to a solar cycle.

We changed the sentence to:

"*Furthermore, the life span of a single satellite instrument is generally shorter than (comparable to in some cases, e.g. MIPAS-ENVISAT, MLS-Aura) one solar cycle*"

2. (p. 3, l. 24) Why "nonlinear"? Any dynamics will affect ozone.

The term nonlinear has been removed.

3. (p. 7) The description of simulated solar flux variation was confusing. You imply that the variations are included in your photolysis lookup table but I could not tell exactly how. Is the table recalculated every day? Is there a separate table for each value of your solar flux parameter? Please be explicit. Also, the impact of the solar variation on heating rate is not clear. Reading between the lines, I guess what you mean is that the heating will respond to the increased or decreased ozone but that, in the heating part of the calculation, the solar flux is kept constant. Is that what you mean? And one other comment: O2 should also be included on your list of radiatively active gases.

We now provide more details on the fact that the photolysis separate look-up table is calculated every day from daily NRLSSI solar flux. We added the following sentence:

"*A separate photolysis look-up table is calculated every day using the daily NRLSSI as solar input*"

We also now provide more explanation on the effect of neglecting UV variations associated with the 27-day rotational cycle on heating rates. The following paragraph has been added:

"*Note however that the direct effect on heating rates generated by UV variations associated with the 27-day rotational cycle is neglected: i.e. daily changes in the spectral irradiance are not considered in the CCM radiative scheme. As a consequence, part of the thermal and dynamical responses to the 27-day rotational cycle and hence their effect on ozone (through transport and temperature dependent chemical reactions, as described above) are missing. The impact of this approximation on our results seems to be small though, as discussed thereafter (sections 3 and 5). Note also that on timescales of the 11yr cycle, Swartz et al. (2012) found that their photolysis-only simulation captured almost all of the solar cycle effect on ozone.*"

Finally, we added $O_2$ in the list of radiatively active gases.

4. (Figure 3) There is a mismatch between the level shown in the figure (~3 hPa) and the level where you see a response in Figure 4 (~10 hPa). Perhaps this is related the mismatch between the SAGE/SBUV results, which contributed to the conclusions in the Hood paper you cite to choose the level of maximum response, and other data and models (e.g. see discussion by Dhomse et al., 2016). Also, a better label for the area where you see a signal would be middle stratosphere, not upper.

The pressure level at ~3 hPa level is the one we used throughout the manuscript as it corresponds to the level where the maximum sensitivity response is found (not only in our study but also in many other observational and model based study). Note that we repeated the power spectrum analysis at 10 hPa and similar conclusions were reached. This has been now clarified in the main text:

"*The analysis has been repeated at lower pressure-height levels (e.g. 10 hPa, not shown) and led to the same conclusions.*"

Regarding the fact that the stronger coherency signal is found at 10 hPa while the maximum sensitivity is found at 3 hPa is explained in the detail in the answer to major comment 6.

5. (p. 9, l. 9) "This explains why ..." This could be the explanation but, as you show later, you are not using enough data to determine a robust signal. It would be safer to say that "This could contribute etc.". Since you cannot see a signal in the observational analysis, it is not appropriate to assume that the response is there but the signal is masked. There may not be a response.

We now changed the text as follows:

"*This illustrates the difficulty in detecting solar rotational signals in the observations, as well as in a single ensemble member over these 3year periods.*"

6. (p. 10, l. 28) "The absence of correlation signal in the middle and lower stratosphere in the observations is consistent with the large noise present in the ozone dataset at these altitudes" As in the comment above, this is misleading since it implies that there is an ozone response but it is masked. You have not shown that a response exists.

There is a signal found in the CTM in the lower stratosphere (below 10 hPa). This signal is however not found in observations. We reformulated the text as follows:

"*The fact that the correlation signal in the middle and lower stratosphere (below 10 hPa) is found in the CTM but not in the observations may partly arise from the large noise present in the MLS-UARS ozone dataset at these altitudes (not shown)*".

7. (p. 13, l. 10) "overall anti-correlation" All I see is that there is a period when F205 variance is low and sensitivity variance is high. The curves are otherwise not related and, even in this period, do not follow a similar evolution. Coincidence of one perturbation is not enough to deduce anti-correlation.

We reformulated as follows:

*"This is further supported by the apparent inverse relationship which is found between the F205 index variance (Fig. 8b) and the ozone sensitivity variance (Fig. 8d)"*

**Editorial comment**

"increasing (decreasing)" and similar construction is grammatically incorrect and very confusing, especially since elsewhere you use parentheses in their legitimate use to define or clarify, e.g. "solar forcing index (F205)".

We modified the following sentences:

[revised manuscript text omitted]

---

## Referee Report (RR1)

**Review of ACPD MS acp-2016-1102-revised**

"Sensitivity of the tropical stratospheric ozone response to the solar rotational cycle in observations and chemistry-climate model simulations" by R. Thiéblemont et al.

I have had time now to read through the revised manuscript and the responses to my earlier review. Publication can now be recommended after revision to make the following minor but important (in my opinion) remaining modifications.

(1) While the statement made in section 2.3 (p. 7) of the first manuscript ("We do not take into account the direct effect on heating rates generated by UV variations because previous modelling studies have already shown that the stratospheric ozone response to solar variations is almost entirely driven by the effects of UV changes on the photolysis rates, in particular the photolysis of molecular oxygen (Swartz et al, 2012).") has thankfully been removed, it has been replaced with the following statement (p. 8, lines 7-8 of the new manuscript): "Note also that on timescales of the 11vr cycle, Swartz et al. (2012) found that their photolysis-only simulation captured almost all of the solar cycle effect on ozone." This statement is also untrue in the upper stratosphere. It appears to be based on a statement in the Swartz et al. paper (p. 5942, bottom of first column): "The photolysis-only simulation captures almost all of the solar cycle effect on ozone." However, these authors were probably referring in this sentence only to the ozone response in the lower stratosphere (below 10 hPa). Looking at their Figure 2, it is clear that the statement is not true in the middle and upper stratosphere, which is the main area of interest here. As mentioned in my first review, at 2 hPa, the ozone response is reduced by the direct heating effect by about 30%. At 1 hPa, the reduction is about 40%. So, please remove this sentence as it will lead to confusion on the part of readers who may be led to believe that temperature feedback effects on the ozone response can be neglected on all time scales.

(2) Despite the comparisons presented with the results of Sukhodolov et al. (2017), who used a CCM that included the direct effect of solar UV variations on the heating rates and the temperature response, it remains unclear to this reviewer that this component of the temperature response can be neglected when calculating the ozone response on the 27-day time scale. Although it appears that increased hydroxyl production is the main cause of the negative ozone phase lags in the present model simulations in the upper stratosphere, is the same true in other models (such as that employed by Sukhodolov et al.)? Models can differ in their photolysis and chemistry schemes as well as their radiation schemes. The only way to investigate this is to improve the radiation scheme of the LMDz-Reprobus CCM and carry out sensitivity studies using both CCMs. Fortunately, there is already a statement in the summary section of the revised manuscript (p. 17, lines 26-27): "Nonetheless, we recognize that to quantify properly the impact of the neglected solar-induced temperature feedback on our results, additional CCM experiments including the direct radiative effect of UV on heating rate should be performed." What is missing in this last section, however, is a statement that such an improvement is necessary for ultimately applying the model to investigate the top-down sun-climate mechanism (noted in the Introduction), which is based on dynamical consequences of the upper stratospheric thermal response. There are apparently quite significant dynamical responses on the 27-day time scale that can have non-negligible tropospheric consequences (e.g., Hood [GRL, v. 43, p. 4066, 2016]). Any hope of simulating the latter effects will depend on an accurate simulation of the direct UV forcing in the upper stratosphere. So please add such a statement.

---

## Author Response (AR2)

We thank the reviewer for reading the manuscript one more time and providing helpful comments and suggestions. We address the raised issues in turn below.

**Our answers are written in blue and manuscript changes are written in italic type within double quotes ("*like this*").**

Answering my first major comment the authors state that neglecting the direct heating rate response to solar irradiance variability does not play substantial role. In support of this statement they use the results published by Sukhodolov et al. (2017) and Swartz et al. (2012). I do not find these arguments convincing. Swartz et al. (2012. Figure 2) show the results averaged over 60 deg. South to 60 deg. North. It is known that the temperature response to solar irradiance variability maximizes in the tropical area and therefore these results do not confirm the author's hypothesis about the small influence of the pure radiative heating. I guess, for the tropical area (which is considered in the manuscript) the contribution of the direct radiative heating response should be 2-3 times higher than 10% obtained by Swartz et al. (2012). It is partially confirmed by Sukhodolov et al. (2016, fig. 4, doi:10.1002/ 2015JD024277) which shows about 28% reduction of the ozone response by direct heating rate responses. I do not think that 28% can be qualified as small and negligible effect. The comparison with the results of Sukhodolov et al., (2017) is also not convincing, because we cannot expect identical results from two models using completely different chemical and photolysis modules. It was shown by Sukhodolov et al. (2016, Figure 7) that the photolysis rates calculated with SOCOL and LMDZ-Reprobus can differ quit substantially. It could well be that the same ozone response in two models is the result of higher pure photochemical response in SOCOL compensated by the inclusion of direct heating rate response. If it is the case the reported in the manuscript O3 response is overestimated by about 30%. I think, the authors should agree on this and describe it properly in the text. It will be better than the use of wrong arguments to defend the applied experimental set-up.

We thank the reviewer for this comment and agree that the comparison with Sukhodolov et al. [2017] - though it is relevant - does not provide a conclusive evidence about the lack of importance of the temperature feedback on ozone. At this time scale, we still think that the ozone response is dominated by the changes in photochemistry. However, we agree that the only way to quantify it is to perform two sets of numerical simulations (a set with solar-driven changes in chemistry versus a set with solar-driven changes in both chemistry and in solar inputs to the radiative scheme). As recommended by the reviewer, we have rephrased the main text in order to:

- discuss in more details what could be the effect of neglecting the direct heating rates on the ozone response (we particularly refer to the results of Sukhodolov et al. (2016)). We however add the caveat that the results in Sukhodolov et al. (2016) are obtained with 1-D RCPM and considering the 11-year. They may change when considering 27-day time scales and when the dynamics are accounted for.
- clarify that, although Sukhodolov et al. (2017) results compare well with ours, models are different and may agree fortuitously. This comparison is relevant to mention, but indeed does not prove that the direct heating rate effect on ozone is much smaller than the direct chemistry effect at 27-day time scales.
- express the need of (i) improving the radiative scheme of LMDz-Reprobus and (ii) performing new experiments to test, at 27-day time scales, the impact of this feedback.

The following changes to the manuscript have been made:

- At various places in the manuscript (sections 2.3, 3.2, 5) we have removed statements on the fact that neglecting the direct heating effect has a limited impact on the results.
- In section 3.2:

"Above 3 hPa (~40 km), CCM cross-correlations of both periods (Fig. 6c,f) show a maximum at negative time lag (-2 days). As mentioned in the introduction, this negative time lag can be induced by temperature feedback on ozone and by increasing hydrogen radical HOx from enhanced solar irradiance which contributes to ozone destruction. While our model configuration allows to fully account for the HOx effect, the solar-induced temperature response is limited since the direct radiative heating effect is not included in the model. The temperature response to the 27-days cycle is thus solely controlled by the ozone concentration change (caused by photolysis changes) and not from the direct heating effect driven by solar irradiance change. Although a temperature signal is found (not shown), it is small, hence reducing the likelihood for the solar-induced temperature feedback to be prominent in our experiments. It is interesting to notice that the upper stratosphere negative lags in our experiments compare very well with those found in CCM experiments of Sukhodolov et al. (2017) (see their Fig. 3) despite the fact that their model (SOCOL) also includes the direct radiative heating effect. At first glance, this good agreement with our model results may suggest that neglecting the direct effect on heating rates generated by UV variations has a limited effect on the ozone response, at least at 27-day timescales. However, this conclusion cannot be drawn because the two models have different photolysis, chemistry and radiation schemes. In particular, it has been shown recently that the photolysis rates calculated by LMDZ-Reprobus and SOCOL can differ substantially (Sukhodolov et al., 2016). The good correspondence between the two sets of model results may thus be fortuitous. For instance, the difference in the photochemical response between SOCOL and LMDz-Reprobus could be compensated by the direct heating rate effect included in SOCOL. Also, a better evaluation of the impact of the direct radiative heating effect requires to perform LMDz-Reprobus experiments, with an increased spectral resolution of the radiative scheme, which account for daily fluctuations of the SSI."

• In section 5:

"In our CCM experimental design, the direct radiative effect of UV on heating rates has been neglected leading to an underestimated temperature response to the 27day cycle. As a consequence, this may affect the ozone response significantly by reducing the temperature feedback on chemical reaction rates, notably ozone destruction through the Chapman cycle. Recently, Sukhodolov et al. (2016) examined the separate effects of heating rates and photolysis rates in solar-driven ozone changes using a 1D radiative-convective-photochemical model and different SSI datasets. Using the NRLSSI solar forcing dataset, they showed that, over the course of the 11-year solar cycle, the direct heating rate anomaly leads to a decrease in ozone of 1% in the middle and upper stratosphere (above 30 hPa) while the photolysis induces an ozone increase of 2 to 4%. Since, the direct radiative effect of UV on heating rates is neglected in our CCM experiments, the ozone response to solar variability may hence be overestimated. Nevertheless, a comparison of the ozone response in our analysis with results from previous independent CCM studies (Rozanov et al., 2006 ; Sukhodolov et al., 2017) revealed a very good correspondence, despite the fact that their experimental design included the direct radiative heating effect. This comparison must be considered with caution as Sukhodolov et al. (2016) found substantial differences in calculated photolysis rates between LMDz-Reprobus and SOCOL photolysis codes. Therefore, accounting for the direct heating rate effect in SOCOL may compensate differences between the two models in ozone response controlled by photochemical processes only. In addition, the results of Sukhodolov et al. (2016) are based on 1-D model calculations and may also change when accounting for dynamical variability (i.e. using 3-D CCM), particularly at 27day time scales where the atmospheric internal variability largely dominates stratospheric temperature variability (Sukhodolov et al., 2017).

To quantify the impact of neglecting solar-induced temperature feedback on our results, the spectral resolution of the LMDz-Reprobus radiative scheme should also be increased and new experiments including the direct radiative effect of UV on heating rate should be performed. We further notice that these improvements are necessary to simulate the "top-down" mechanism which is based on dynamical consequences of the upper stratospheric thermal response."

**Anonymous Referee #2**

Review of ACPD MS acp-2016-1102-revised "Sensitivity of the tropical stratospheric ozone response to the solar rotational cycle in observations and chemistry-climate model simulations" by R. Thiéblemont et al.

I have had time now to read through the revised manuscript and the responses to my earlier review. Publication can now be recommended after revision to make the following minor but important (in my opinion) remaining modifications.

We thank the reviewer for reading the manuscript one more time and providing helpful comments and suggestions. We address the raised issues in turn below.

Our answers are written in blue and manuscript changes are written in italic type within double quotes (*"like this"*).

(1) While the statement made in section 2.3 (p. 7) of the first manuscript ("We do not take into account the direct effect on heating rates generated by UV variations because previous modelling studies have already shown that the stratospheric ozone response to solar variations is almost entirely driven by the effects of UV changes on the photolysis rates, in particular the photolysis of molecular oxygen (Swartz et al, 2012).") has thankfully been removed, it has been replaced with the following statement (p. 8, lines 7-8 of the new manuscript): "Note also that on timescales of the 11yr cycle, Swartz et al. (2012) found that their photolysis-only simulation captured almost all of the solar cycle effect on ozone." This statement is also untrue in the upper stratosphere. It appears to be based on a statement in the Swartz et al. paper (p. 5942, bottom of first column): "The photolysis-only simulation captures almost all of the solar cycle effect on ozone." However, these authors were probably referring in this sentence only to the ozone response in the lower stratosphere (below 10 hPa). Looking at their Figure 2, it is clear that the statement is not true in the middle and upper stratosphere, which is the main area of interest here. As mentioned in my first review, at 2 hPa, the ozone response is reduced by the direct heating effect by about 30%. At 1 hPa, the reduction is about 40%. So, please remove this sentence as it will lead to confusion on the part of readers who may be led to believe that temperature feedback effects on the ozone response can be neglected on all time scales.

**We agree with the reviewer and the sentence has been removed. We also now discussed more thoroughly the neglect of the direct heating rates on ozone, as requested also by reviewer #1. See our answer to the comment 2) for more details.**

(2) Despite the comparisons presented with the results of Sukhodolov et al. (2017), who used a CCM that included the direct effect of solar UV variations on the heating rates and the temperature response, it remains unclear to this reviewer that this component of the temperature response can be neglected when calculating the ozone response on the 27-day time scale. Although it appears that increased hydroxyl production is the main cause of the negative ozone phase lags in the present model simulations in the upper stratosphere, is the same true in other models (such as that employed by Sukhodolov et al.)? Models can differ in their photolysis and chemistry schemes as well as their radiation schemes. The only way to investigate this is to improve the radiation scheme of the LMDz-Reprobus CCM and carry out sensitivity studies using both CCMs. Fortunately, there is already a statement in the summary section of the revised manuscript (p. 17, lines 26-27): "Nonetheless, we recognize that to quantify properly the impact of the neglected solar-induced temperature feedback on our results, additional CCM experiments including the direct radiative effect of UV on heating rate should be performed." What is missing in this last section, however, is a statement that such an improvement is necessary for ultimately applying the model to investigate the top-down sun-climate

mechanism (noted in the Introduction), which is based on dynamical consequences of the upper stratospheric thermal response. There are apparently quite significant dynamical responses on the 27-day time scale that can have non-negligible tropospheric consequences (e.g., Hood [GRL, v. 43, p. 4066, 2016]). Any hope of simulating the latter effects will depend on an accurate simulation of the direct UV forcing in the upper stratosphere. So please add such a statement.

We thank the reviewer for this comment and agree that the comparison with Sukhodolov et al. [2017] - though it is relevant - does not constitute a sufficient evidence to exclude the effect of the temperature feedback on ozone. This critical was also expressed by Reviewer 1. We revised the manuscript in order to:

- discuss in more details what could be the effect of neglecting the direct heating rates on the ozone response (we particularly refer to the results of Sukhodolov et al. (2016)). We however add the caveat that the results in Sukhodolov et al. (2016) are obtained with 1-D RCPM and considering the 11-year. They may change when considering 27-day time scales and when the dynamics are accounted for.
- clarify that, although Sukhodolov et al. [2017] results compare well with ours, models are different and may agree fortuitously. This comparison is relevant to mention, but indeed does not prove that the direct heating rate effect much smaller than the direct chemistry effect at 27-day timescales.
- express the need of (i) improving the radiative scheme of LMDz-Reprobus and (ii) performing new experiments to test, at 27-day time scales, the impact of this feedback.

As requested, we also added a statement about future improvement (currently under development) of the LMDz-Reprobus model that are necessary for simulating the "top-down" mechanism.

The following changes to the manuscript have been made:

**In section 3.2:**

"Above 3 hPa (~40 km), CCM cross-correlations of both periods (Fig. 6c,f) show a maximum at negative time lag (-2 days). As mentioned in the introduction, this negative time lag can be induced by temperature feedback on ozone and by increasing hydrogen radical HOx from enhanced solar irradiance which contributes to ozone destruction. While our model configuration allows to fully account for the HOx effect, the solar-induced temperature response is limited since the direct radiative heating effect is not included in the model. The temperature response to the 27-days cycle is thus solely controlled by the ozone concentration change (caused by photolysis changes) and not from the direct heating effect driven by solar irradiance change. Although a temperature signal is found (not shown), it is small, hence reducing the likelihood for the solar-induced temperature feedback to be prominent in our experiments. It is interesting to notice that the upper stratosphere negative lags in our experiments compare very well with those found in CCM experiments of Sukhodolov et al. (2017) (see their Fig. 3) despite the fact that their model (SOCOL) also includes the direct radiative heating effect. At first glance, this good agreement with our model results may suggest that neglecting the direct effect on heating rates generated by UV variations has a limited effect on the ozone response, at least at 27-day timescales. However, this conclusion cannot be drawn because the two models have different photolysis, chemistry and radiation schemes. In particular, it has been shown recently that the photolysis rates calculated by LMDZ-Reprobus and SOCOL can differ substantially (Sukhodolov et al., 2016). The good correspondence between the two sets of model results may thus be fortuitous. For instance, the difference in the photochemical response between SOCOL and LMDz-Reprobus could be compensated by the direct heating rate effect included in SOCOL. Also, a better evaluation of the impact of the direct radiative heating effect

requires to perform LMDz-Reprobus experiments, with an increased spectral resolution of the radiative scheme, which account for daily fluctuations of the SSI."

• In section 5:

"In our CCM experimental design, the direct radiative effect of UV on heating rates has been neglected leading to an underestimated temperature response to the 27day cycle. As a consequence, this may affect the ozone response significantly by reducing the temperature feedback on chemical reaction rates, notably ozone destruction through the Chapman cycle. Recently, Sukhodolov et al. (2016) examined the separate effects of heating rates and photolysis rates in solar-driven ozone changes using a 1D radiative-convective-photochemical model and different SSI datasets. Using the NRLSSI solar forcing dataset, they showed that, over the course of the 11-year solar cycle, the direct heating rate anomaly leads to a decrease in ozone of 1% in the middle and upper stratosphere (above 30 hPa) while the photolysis induces an ozone increase of 2 to 4%. Since, the direct radiative effect of UV on heating rates is neglected in our CCM experiments, the ozone response to solar variability may hence be overestimated. Nevertheless, a comparison of the ozone response in our analysis with results from previous independent CCM studies (Rozanov et al., 2006; Sukhodolov et al., 2017) revealed a very good correspondence, despite the fact that their experimental design included the direct radiative heating effect. This comparison must be considered with caution as Sukhodolov et al. (2016) found substantial differences in calculated photolysis rates between LMDz-Reprobus and SOCOL photolysis codes. Therefore, accounting for the direct heating rate effect in SOCOL may compensate differences between the two models in ozone response controlled by photochemical processes only. In addition, the results of Sukhodolov et al. (2016) are based on 1-D model calculations and may also change when accounting for dynamical variability (i.e. using 3-D CCM), particularly at 27day time scales where the atmospheric internal variability largely dominates stratospheric temperature variability (Sukhodolov et al., 2017). To quantify the impact of neglecting solar-induced temperature feedback on our results, the spectral resolution of the LMDz-Reprobus radiative scheme should also be increased and new experiments including the direct radiative effect of UV on heating rate should be performed. We further notice that these improvements are necessary to simulate the "top-down" mechanism which is based on dynamical consequences of the upper stratospheric thermal response."

**Anonymous Referee #3**

The authors have extensively revised the paper in response to my comments and those from the other reviewers. The paper is acceptable after minor revisions.

We thank the reviewer for reading the manuscript one more time and providing helpful comments and suggestions. We address the raised issues in turn below.

Our answers are written in blue and manuscript changes are written in italic type within double quotes (*"like this"*).

The results seem to be reliable and well-explained within the context of the simulations performed and therefore are publishable. However, I am still concerned that the authors have chosen to go ahead with their analysis even though they acknowledge the shortcoming of the model due to the neglect of daily solar flux variability in the heating calculation. This is laid out quite in the open so the reader is not being misled in any way. There will no doubt be some readers who mistrust the results since they are derived from simulations using a model that is not exactly appropriate for this particular investigation. As far as I can tell, the reason for omitting this key physical process is not that it is difficult to include because of technical issues or a poor understanding of the physics but rather because they prefer not to rerun the model with the correction. Developing model routines and running for long simulations take a lot of time, effort, and computing resources. That effort and expense can be worthwhile if and when it gives confidence that the results represent the current understanding of the atmosphere.

**Minor comments:**

(p. 2; l. 23) You could also add Envisat GOMOS and TIMED SABER to your list.

**Done**

(p. 4, l. 12-15) Check the logic of these two sentences; Gruzdev (misspelled in the manuscript) et al -> enhanced solar gives reduced O3 sensitivity but then Kubin et al -> weak solar gives enhanced O3 sensitivity. That's the same, right?

Indeed, it is the same, that is why we used the term "reciprocally". The two sentences seem logic to us.

**We corrected the Gruzdev spelling at two places.**

(p. 12, l. 3) It would be more accurate to say that the temperature response is controlled by the ozone concentration, rather than the ozone production. The solar flux affects both production and loss of ozone.

**We corrected it. Thank you for noticing this.**

(p. 12, l. 9-10) ("More sensitivity experiments") Related to the general comment above, it would be a better use of your time to incorporate a heating routine that includes the option of daily varying spectrally resolved solar flux. More sensitivity experiments are not recommended; now is a good time to move on to an updated algorithm.

We agree with the reviewer and modified the sentence as follows:

"Also, a better evaluation of the impact of the direct radiative heating effect requires to perform LMDz-Reprobus experiments, with an increased spectral resolution of the radiative scheme, which account for daily fluctuations of the SSI."

(p. 12, l. 29) "errors" This is not the right word since you are showing variations that include other physical processes such as dynamics and do not include systematic errors.

**We change it to "uncertainties of the sensitivity profile estimates"**

(p. 15, discussion of Figure 9) I still have trouble seeing the inverse relationships that are used to motivate the later analysis (9b vs. 9d; 9b vs. 9c). Could you add an extra panel to Figure 9 that would overlay these so that the inverse relationships are more apparent?

In our opinion, adding an extra panel is not necessary since this is more or less what is shown on Figure 10. However, we recognize that the tone could be less affirmative regarding the inverse relationship (since it turns out that it is not robust regarding the mean ozone sensitivity). We thus modified the text as follows:

"Finally, note that the low-frequency (i.e. decadal scales) variability of the ensemble mean ozone sensitivity (Fig. 9c) may also suggest an inverse relationship with the F205 absolute value (Fig. 9a) and its variance (Fig. 9b). In the following, we investigate further the relationships suggested here which link the solar rotational variability to the ensemble mean and spread of ozone sensitivity."

**Editorial comments:**

**(p. 8, l. 29) What is the subject of "cover"?**

We corrected to "*covers*" since "solar rotational cycle" is the subject. Thanks for noticing the mistake.

**(p. 9, l. 26) What does "presumably because" refer to?**

It refers the fact that the peak is less pronounced.

We split the sentence in two to make the point clearer.

"For 2004-07, the peak is centred at 25 days (Fig. 3f). The peak is also less pronounced than in 1991-94, presumably because of the smaller amplitude of solar rotational fluctuations and hence model forcing in 2004-07 (see Fig. 2)"

**(p. 11; l. 11-13) This sentence has no verb.**

We corrected "result" for "results" (the subject is "negative lag")

**(p. 11, l. 28) "difference" from what?**

We changed "difference" for "discrepancies"

**(caption to Figure 7) The last sentence is repeated.**

The first sentence refers to regression estimates error while the second refers to model ensemble spread. This is now clarified.

**(p. 15, l. 32) "shown in the insert of Fig. 9a" -> "shown in Fig. 9a"**

Done

**Sensitivity of the tropical stratospheric ozone response to the solar rotational cycle in observations and chemistry-climate model simulations**

Rémi Thiéblemont1, Marion Marchand1, Slimane Bekki1, Sébastien Bossay1, Franck Lefèvre1, Mustapha Meftah1, Alain Hauchecorne1

[revised manuscript text omitted]